# MedCRP-CL: Continual Medical Image Segmentation via Bayesian Nonparametric Semantic Modality Discovery

**Ziyuan Gao** [1]

## Abstract

Medical image segmentation faces a fundamental challenge in continual learning: data arrives sequentially from heterogeneous sources, yet effective continual learning requires discovering which tasks share sufficient structure to benefit from joint learning. Existing methods either apply uniform constraints across all tasks, causing catastrophic forgetting when tasks conflict, or require predefined task groupings that cannot anticipate future task diversity. We introduce MedCRP-CL, a framework that performs online task structure discovery and structure-aware continual learning. Leveraging the Chinese Restaurant Process (CRP), our method dynamically infers task groupings from clinical text prompts as tasks arrive, without requiring predefined cluster counts or access to future tasks. We term these discovered groupings semantic modalities, as they capture finer-grained structure than physical imaging modalities by integrating anatomical region and pathological context. Guided by this discovered structure, we maintain semantic modality-specific LoRA adapters regularized by intra-modality EWC, ensuring parameter isolation across dissimilar task groups while facilitating knowledge transfer within similar ones. The framework is also replay-free, storing only aggregate statistics rather than raw patient data. Experiments on 16 medical segmentation tasks across four imaging modalities demonstrate that MedCRP-CL achieves 73.3% Dice score with only 4.1% forgetting, outperforming the best baseline by 8.0% while requiring 6× fewer parameters. Code is available at https://github.com/zygao930/MedCRP-CL.

[1]University College London, London, United Kingdom. Correspondence to: Ziyuan Gao <ucbqzg5@ucl.ac.uk>.

*Proceedings of the $43^{rd}$ International Conference on Machine Learning*, Seoul, South Korea. PMLR 306, 2026. Copyright 2026 by the author(s).

## 1. Introduction

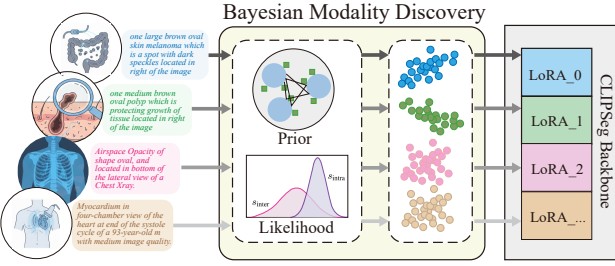

*Figure 1.* Overview of MedCRP-CL. Clinical prompts are assigned to semantic modalities via Bayesian inference combining CRP prior and adaptive likelihood.

Medical image segmentation is fundamental to clinical diagnosis and treatment planning, enabling quantitative analysis of anatomical structures and pathological regions (Poudel et al., 2024). In clinical practice, medical imaging data is routinely aggregated from heterogeneous sources including public benchmarks, multi-center trials, and partner institutions (Guan & Liu, 2022; Guan et al., 2024; Chang et al., 2023). This data arrives continuously from diverse clinical workflows: chest X-rays from emergency departments, ultrasound scans from cardiology, and endoscopic images from gastroenterology (Perkonigg et al., 2021). Such sequential data acquisition creates a critical challenge for deployed segmentation models: they must continually adapt to new tasks while preserving performance on previously learned ones (Verma et al., 2023; Kumari et al., 2025). Vision-language segmentation models offer promising capabilities for this setting by leveraging natural language prompts to guide segmentation (Lüddecke & Ecker, 2022; Zhao et al., 2025), yet their deployment in continual learning scenarios remains largely unexplored.

Existing continual learning methods face a fundamental tension between parameter sharing and parameter isolation. Methods like EWC (Kirkpatrick et al., 2017) apply uniform parameter constraints across all tasks, forcing the model to compromise between conflicting objectives when tasks are dissimilar (Perkonigg et al., 2021; McCloskey & Cohen, 1989). Mixture-of-experts approaches (Yu et al., 2024) attempt task-specific adaptation but require predefined ex-

pert counts that cannot anticipate future task diversity. The core issue is that indiscriminate sharing accelerates forgetting across dissimilar tasks, while rigid isolation precludes beneficial transfer among related tasks. Addressing this tension requires discovering the underlying task structure: which tasks are sufficiently similar to benefit from shared representations, and which require separate parameters.

Discovering such task structure in medical imaging is non-trivial. Physical imaging modality labels (e.g., "ultrasound", "X-ray"), though readily available in clinical metadata, provide insufficient granularity: cardiac ultrasound and breast ultrasound share the same acquisition principles but involve fundamentally different anatomical structures and pathological patterns (Leclerc et al., 2019; Al-Dhabyani et al., 2020). Clustering based on raw image features is computationally expensive and unreliable due to high dimensionality and cross-site acquisition variability (Zhang et al., 2022). We observe that clinical text prompts offer a more effective basis for task grouping, as they naturally encode the combination of anatomical region and pathological context (Huang et al., 2021; Zhao et al., 2025). For instance, "chest X-ray showing pleural effusion" and "ultrasound of breast lesion" occupy distinct regions in prompt embedding space, reflecting their clinically meaningful differences.

Based on this observation, we present MedCRP-CL, a framework that discovers task groupings from clinical prompt embeddings to guide continual learning (Figure 1). We term these discovered groupings *semantic modalities* to distinguish them from physical imaging modalities, as they capture finer-grained structure combining anatomy and pathology. Given a sequence of tasks $\mathcal{T} = \{T_1, T_2, \ldots, T_N\}$, our framework jointly learns: (1) a clustering function $z : \mathcal{T} \to \mathbb{N}$ assigning tasks to semantic modalities based on prompt embeddings, and (2) semantic modality-specific parameters $\theta_k$ enabling knowledge sharing within modalities while maintaining isolation across them. We employ the Chinese Restaurant Process (Blei et al., 2010) to automatically determine the number of semantic modalities, combined with semantic modality-specific LoRA adapters (Hu et al., 2022) regularized by intra-modality EWC.

Our contributions are threefold:

- We propose a **Bayesian nonparametric approach** for automatic semantic modality discovery in continual medical image segmentation. The CRP-based mechanism dynamically infers task groupings from clinical text prompts without requiring predefined labels or expert counts.

- We design a **semantic modality-aware continual learning architecture** that combines modality-specific LoRA adapters with intra-modality EWC regularization, achieving parameter isolation across semantic

modalities while enabling knowledge transfer within them. Our approach is also **replay-free**, eliminating the need to store historical patient data.

- We demonstrate strong performance on 16 medical segmentation tasks spanning four imaging modalities, outperforming the best baseline by 8.0% in Dice score while achieving only 4.1% forgetting and requiring $6\times$ fewer parameters. The discovered semantic modalities capture **finer-grained structure** than physical imaging modalities alone.

## 2. Related Work

**Continual Learning for Vision-Language Models** Continual learning aims to mitigate catastrophic forgetting during sequential task learning. Regularization-based methods, such as EWC (Kirkpatrick et al., 2017), penalize changes to critical parameters by estimating Fisher information. Replay-based approaches (Rebuffi et al., 2017; Chaudhry et al., 2019) store and rehearse past samples, but are impractical in medical settings where patient data cannot be retained due to privacy regulations such as HIPAA and GDPR. Recently, several works have extended continual learning to vision-language models. RAPF (Huang et al., 2024) combines representation adjustment with parameter fusion. CL-LoRA (He et al., 2025) applies low-rank adaptation alongside knowledge distillation. MoE-Adapters (Yu et al., 2024) employs mixture-of-experts routing for task-specific adaptation. However, these methods often assume homogeneous task distributions or require predefined expert counts, making them suboptimal for medical imaging where tasks involve heterogeneous modalities and unknown structures.

**Continual Medical Image Segmentation** Several recent methods address continual learning for medical image segmentation. MedPEFT-CL (Gao & Morel, 2026) introduces dual-phase parameter-efficient adaptation with bidirectional memory consolidation, but requires a replay buffer that conflicts with clinical privacy constraints. Low-Rank MoE (Chen et al., 2024) allocates an independent LoRA expert per task, avoiding forgetting at the cost of linear parameter growth and no knowledge sharing among semantically related tasks. FR$^2$Seg (Xu et al., 2025) targets cross-site domain adaptation for a fixed segmentation task via Fourier-based transfer, rather than learning diverse tasks sequentially. Recent advances in structure-aware medical imaging such as HFF-Net (Shao et al., 2025b) and TRACE (Shao et al., 2025a) suggest promising directions for extending continual segmentation to 3D volumetric settings.

**Task Structure Discovery in Continual Learning** Existing continual learning methods typically assume tasks are either independent or share a global structure. Dynamic ar-

chitecture methods (Rusu et al., 2016; Yoon et al., 2018) expand network capacity for new tasks but lack mechanisms to identify shared structure. Task-agnostic approaches (Aljundi et al., 2019; Zeno et al., 2021) attempt to detect task boundaries automatically but still treat all tasks uniformly without grouping related ones. CAT (Ke et al., 2020) detects whether new tasks are similar or dissimilar to previous ones and applies different strategies accordingly, but relies on binary classification rather than discovering arbitrary cluster structures. Online clustering methods such as Online K-Means (MacQueen, 1967) and DP-Means (Kulis & Jordan, 2012) can dynamically assign data points to clusters, but require either a predefined number of clusters or sensitive distance thresholds. Bayesian nonparametric methods such as the Chinese Restaurant Process (Blei et al., 2010) offer principled frameworks for clustering with an unknown number of groups, yet their application to continual learning in medical imaging remains unexplored.

## 3. Method

**Problem Formulation**   We consider continual learning for medical image segmentation, where a sequence of tasks $\mathcal{T} = \{T_1, T_2, \ldots, T_N\}$ arrives sequentially from diverse clinical sources. Each task $T_t$ consists of a dataset $\mathcal{D}_t = \{(x_i^t, y_i^t, p_i^t)\}_{i=1}^{n_t}$, where $x_i^t$ denotes a medical image, $y_i^t$ is the segmentation mask, and $p_i^t$ is a clinical text prompt describing the target anatomical structure or pathology.

Medical images are acquired through distinct physical processes that induce fundamentally different visual distributions (Guan & Liu, 2022). We denote this latent structure as semantic modality partition $\mathcal{M} = \{M_1, \ldots, M_K\}$, where $K$ is unknown a priori. Tasks within the same semantic modality share visual characteristics and benefit from parameter sharing, while cross-modality tasks exhibit significant distribution shift that necessitates parameter isolation.

Our objectives are: (1) learn a semantic modality assignment function $z : \mathcal{T} \to \mathbb{N}$ that clusters tasks into semantic modalities without supervision, (2) minimize catastrophic forgetting through semantic modality-aware regularization, and (3) operate without storing raw patient data to comply with privacy regulations. The overall objective is:

$$\mathcal{L} = \sum_{t=1}^{N} \mathcal{L}_{\text{seg}}(T_t; \theta_{z(t)}) + \sum_{k=1}^{K} \Omega_k(\theta_k) \quad (1)$$

where $\theta_{z(t)}$ denotes semantic modality-specific parameters and $\Omega_k$ is the forgetting penalty applied only within semantic modality $k$. Figure 2 illustrates the overall architecture.

### 3.1. Bayesian Nonparametric Modality Discovery

Our framework discovers semantic modalities automatically without predefining their number. Physical modalities are fixed by imaging hardware, but semantic modalities capture finer-grained clinical distinctions that emerge from the data. A Bayesian nonparametric formulation naturally accommodates this: new semantic modalities are instantiated when tasks exhibit novel characteristics, while related tasks consolidate into existing clusters. The discovery process combines a CRP prior over partitions with a likelihood based on prompt embeddings, as detailed below.

**Chinese Restaurant Process Prior**   We employ the Chinese Restaurant Process (CRP) (Blei et al., 2010) as a prior over semantic modality assignments. For task $T_t$, the prior probability is:

$$P(z_t = k \mid z_{1:t-1}, \alpha) = \begin{cases} \frac{n_k}{t-1+\alpha} & k \in \{1, \ldots, K_{t-1}\} \\ \frac{\alpha}{t-1+\alpha} & k = \text{new} \end{cases}$$
$$(2)$$

where $n_k$ counts tasks assigned to semantic modality $k$ and $\alpha > 0$ controls the propensity for new modalities.

The CRP prior is modulated by a likelihood term based on prompt similarity. Even when an existing semantic modality has accumulated many tasks, a new task will only join if its prompt embedding is sufficiently similar to the cluster centroid. This ensures that semantic coherence takes precedence over cluster size in modality assignment.

**Prompt-Based Semantic Similarity**   We extract task representations from clinical prompts using a frozen text encoder $\phi$ from CLIP (Radford et al., 2021). For task $T_t$:

$$e_t = \frac{1}{|\mathcal{P}_t|} \sum_{p \in \mathcal{P}_t} \frac{\phi(p)}{\|\phi(p)\|_2} \quad (3)$$

where $\mathcal{P}_t$ denotes unique prompts in task $t$. Each semantic modality $k$ maintains a running centroid $\mu_k$, and the similarity is computed as $s_{t,k} = \langle e_t, \mu_k \rangle$. When task $t$ is assigned to semantic modality $k$, the centroid is updated online:

$$\mu_k \leftarrow \frac{n_k - 1}{n_k} \mu_k + \frac{1}{n_k} e_t \quad (4)$$

This incremental update computes the exact running mean. Only the centroid vector is stored per semantic modality, avoiding storage of individual task embeddings. The centroid is not renormalized after updates. Its norm implicitly captures cluster coherence, with consistent embeddings maintaining norms close to unity and diverse assignments yielding smaller similarity scores.

**Adaptive Similarity Distributions**   Rather than hand-tuning similarity thresholds, we learn them from data. We model similarity scores as conditionally Gaussian: $s \mid$ same $\sim \mathcal{N}(\mu_{\text{intra}}, \sigma_{\text{intra}}^2)$ for tasks belonging to the same semantic modality, and $s \mid$ diff $\sim \mathcal{N}(\mu_{\text{inter}}, \sigma_{\text{inter}}^2)$ for tasks

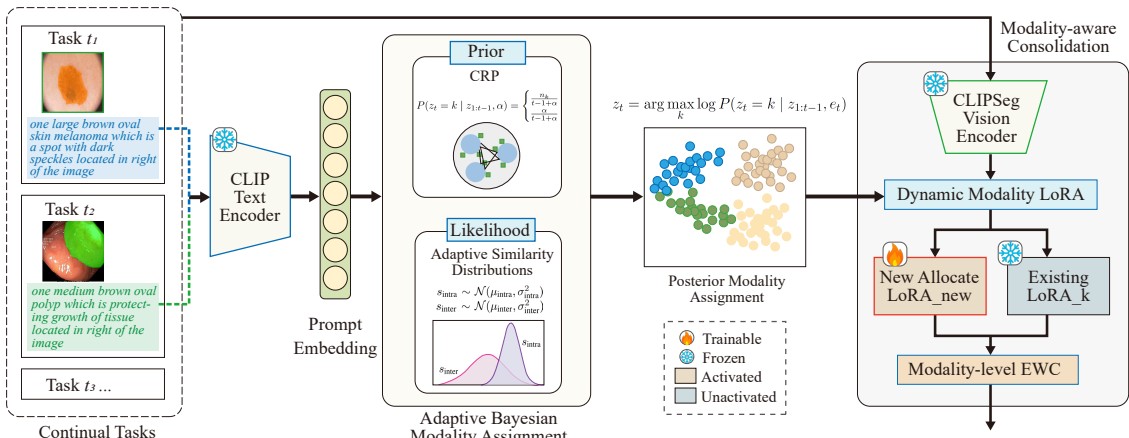

*Figure 2.* Overview of MedCRP-CL. For each incoming task, prompt embeddings are extracted and passed to the Bayesian modality assignment module, which combines CRP prior with learned similarity distributions to determine semantic modality membership. The assigned semantic modality's LoRA adapter is then activated for training with intra-modality EWC regularization.

across different modalities. We maintain online estimates of both distributions, updated via Welford's online algorithm (Welford, 1962). Given an observed similarity $s$ between a new task and an existing semantic modality, this yields the log-likelihood ratio:

$$\ell(s) = \frac{(s - \mu_{\text{inter}})^2}{2\sigma_{\text{inter}}^2} - \frac{(s - \mu_{\text{intra}})^2}{2\sigma_{\text{intra}}^2} + \log \frac{\sigma_{\text{inter}}}{\sigma_{\text{intra}}} \quad (5)$$

To ensure numerical stability, we set a minimum standard deviation $\sigma_{\min} = 0.05$ for both distributions. During cold start when insufficient samples are available (fewer than one observation per distribution), we fall back to a logit-based likelihood $\ell(s) = \log(s + \epsilon) - \log(1 - s + \epsilon)$, treating similarity directly as a probability proxy. The cold-start logit mechanism maintains a similarity margin of 0.22+ between same-modality and cross-modality pairs, ensuring reliable assignment before Gaussian activation (see Appendix E.1). After Gaussian activation, the similarity distributions stabilize rapidly and maintain clear separation throughout training (see Appendix E.2). The distributions are updated as follows: when task $t$ is assigned to modality $k$, the similarity $s_{t,k}$ updates the intra-modality distribution, while similarities $s_{t,j}$ for all $j \neq k$ update the inter-modality distribution. A positive $\ell(s)$ indicates that $s$ is more consistent with intra-modality similarity, favoring assignment to the existing semantic modality.

**Posterior Modality Assignment** Combining prior and likelihood, the posterior for assigning task $t$ to semantic modality $k$ is:

$$\log P(z_t = k \mid z_{1:t-1}, e_t) = \log n_k - \log(t-1+\alpha) + \ell(s_{t,k}) \quad (6)$$

For a new semantic modality, letting $k^* = \arg\max_k s_{t,k}$:

$$\log P(z_t = \text{new}) = \log \alpha - \log(t-1+\alpha) - \ell(s_{t,k^*}) \quad (7)$$

The negative sign reflects that if the new task is dissimilar to all existing modalities (low $\ell(s_{t,k^*})$), it should form a new cluster. We perform MAP inference:

$$z_t = \arg\max_k \log P(z_t = k \mid z_{1:t-1}, e_t) \quad (8)$$

The likelihood term $\ell(s_{t,k})$ ensures that semantic similarity determines assignments. A task joins an existing semantic modality only when its prompt embedding is close to the cluster centroid; otherwise, a new semantic modality is created regardless of existing cluster sizes. Proposition A.1 guarantees near-zero assignment error under mild separation conditions (Appendix A).

### 3.2. Semantic Modality-Specific Continual Learning

Given semantic modality assignments, we instantiate semantic modality-specific parameters. Parameters are isolated across semantic modalities but shared among tasks within the same semantic modality.

**Dynamic Semantic Modality-Specific LoRA** We build upon a frozen vision-language backbone $f_\Theta$ (CLIPSeg) with semantic modality-specific low-rank adapters. For each linear layer $W_0 \in \mathbb{R}^{d_{\text{out}} \times d_{\text{in}}}$, semantic modality $k$ maintains:

$$W_k = W_0 + \frac{\alpha_{\text{LoRA}}}{r} B_k A_k \quad (9)$$

where $A_k \in \mathbb{R}^{r \times d_{\text{in}}}$, $B_k \in \mathbb{R}^{d_{\text{out}} \times r}$, and $r \ll d$. When a new semantic modality is discovered, fresh $(A_k, B_k)$ pairs are allocated. This design achieves two goals: complete parameter isolation across semantic modalities prevents negative transfer between incompatible imaging types, while parameter sharing within each semantic modality enables positive transfer among related tasks.

**Algorithm 1** MedCRP-CL: Adaptive CRP for Continual Medical Image Segmentation

---

**Input:** $\mathcal{T} = \{T_t\}_{t=1}^N$: Task sequence; $\phi(\cdot)$: Text encoder; $\alpha$: CRP concentration; $\lambda$: EWC coefficient
**Output:** $\{\theta_k\}_{k=1}^K$: Semantic modality-specific LoRA parameters
Initialize $\mathcal{M} \leftarrow \emptyset$; $(\mu_{\text{intra}}, \sigma_{\text{intra}}, \mu_{\text{inter}}, \sigma_{\text{inter}}) \leftarrow (0, 1, 0, 1)$
**for** each task $T_t = (\mathcal{D}_t, \mathcal{P}_t)$ in $\mathcal{T}$ **do**
$\quad \triangleright$ Semantic Modality Discovery
$\quad$ Extract embedding $e_t \leftarrow \mathbb{E}_{p \sim \mathcal{P}_t}[\phi(p)/\|\phi(p)\|]$
$\quad$ **for** each semantic modality $k$ in $\mathcal{M}$ **do**
$\quad\quad$ Compute similarity $s_{t,k} \leftarrow \langle e_t, \mu_k \rangle$
$\quad\quad \log P_k \leftarrow \log \frac{n_k}{t-1+\alpha} + \ell(s_{t,k})$
$\quad$ **end for**
$\quad \log P_{\text{new}} \leftarrow \log \frac{\alpha}{t-1+\alpha} - \ell(\max_k s_{t,k})$
$\quad z_t \leftarrow \arg\max \log P$; allocate new LoRA if $z_t = $ new
$\quad \triangleright$ Modality-Specific Training
$\quad \mathcal{L} \leftarrow \mathcal{L}_{\text{CE}} + \mathcal{L}_{\text{Dice}} + \mathbf{1}_{[n_{z_t}>1]} \cdot \Omega_{z_t}$
$\quad$ Update $\theta_{z_t}$ until convergence
$\quad \triangleright$ Statistics Update
$\quad$ Update $\mu_{z_t}$ and similarity distributions
$\quad$ Compute $F_{z_t}^{(t)}$; update $\bar{F}_{z_t} \leftarrow \frac{n_{z_t}-1}{n_{z_t}}\bar{F}_{z_t} + \frac{1}{n_{z_t}}F_{z_t}^{(t)}$
$\quad$ Store anchor parameters $\theta_{z_t}^* \leftarrow \theta_{z_t}$
**end for**
**Return** $\{\theta_k\}_{k=1}^{|\mathcal{M}|}$

---

**Intra-Modality Elastic Weight Consolidation** Parameter isolation across semantic modalities prevents cross-modality interference, but tasks within the same semantic modality still share parameters and may overwrite each other. We apply Elastic Weight Consolidation (EWC) to prevent this intra-modality forgetting. After training task $t$ in semantic modality $k$, we estimate the Fisher information:

$$F_k^{(t)} = \mathbb{E}_{(x,y)\sim \mathcal{D}_t}\left[\nabla_{\theta_k} \log p(y|x; \theta_k)^{\otimes 2}\right] \quad (10)$$

The consolidated Fisher for semantic modality $k$ is updated via exponential moving average:

$$\bar{F}_k \leftarrow \frac{n_k - 1}{n_k}\bar{F}_k + \frac{1}{n_k}F_k^{(t)} \quad (11)$$

For subsequent tasks in semantic modality $k$, the regularization term is:

$$\Omega_k(\theta_k) = \sum_i \bar{F}_{k,i}(\theta_{k,i} - \theta_{k,i}^*)^2 \quad (12)$$

EWC applies only within semantic modalities. Tasks in one semantic modality have no interaction with parameters of another, preventing conflicting gradients from incompatible visual distributions.

The training objective combines segmentation and regularization losses:

$$\mathcal{L} = \mathcal{L}_{\text{CE}} + \mathcal{L}_{\text{Dice}} + \Omega_{z(t)}(\theta_{z(t)}) \quad (13)$$

where $\mathcal{L}_{\text{CE}}$ is cross-entropy loss, $\mathcal{L}_{\text{Dice}}$ addresses class imbalance in medical images, and $\Omega_{z(t)}$ regularizes only the current semantic modality's parameters. $\Omega_{z(t)} = 0$ for the first task in each semantic modality. Algorithm 1 summarizes the complete procedure.

## 4. Experiments

### 4.1. Experimental Setup

**Datasets** We evaluate on 16 medical image segmentation tasks spanning four imaging types and five anatomical regions: endoscopy (Jha et al., 2020; Bernal et al., 2015; Silva et al., 2014; Tajbakhsh et al., 2016; Vázquez et al., 2017), dermoscopy (Codella et al., 2018), ultrasound (Leclerc et al., 2019; Al-Dhabyani et al., 2020), and chest X-ray (Irvin et al., 2019). Text prompts are adopted from MedVLSM (Poudel et al., 2024). Table 1 summarizes the dataset statistics. Detailed dataset descriptions are provided in Appendix B.

*Table 1.* Dataset statistics for continual medical image segmentation.

| Imaging Type | Organ | Dataset | # Train/Val/Test |
|---|---|---|---|
| Endoscopy | Colon | Kvasir | 800/100/100 |
| | | ClinicDB | 490/61/61 |
| | | ETIS | 137/39/20 |
| | | CVC-300 | 64/21/23 |
| | | ColonDB | 266/76/38 |
| Dermoscopy | Skin | ISIC | 810/90/379 |
| Ultrasound | Heart | CAMUS | 960/120/120 |
| | Breast | BUSI-Benign | 349/44/44 |
| | | BUSI-Malignant | 168/21/21 |
| X-ray | Chest | Airspace Opacity | 75/17/33 |
| | | Atelectasis | 48/9/23 |
| | | Cardiomegaly | 35/14/19 |
| | | Edema | 25/9/11 |
| | | Pleural Effusion | 39/10/18 |
| | | Enlarged Cardio. | 57/18/34 |
| | | Support Devices | 59/17/31 |

**Metrics** We evaluate using two metrics: (1) **Average Dice Coefficient** measuring segmentation accuracy:

$$\text{Avg Dice} = \frac{1}{T}\sum_{i=1}^T \frac{2|P_i \cap G_i|}{|P_i| + |G_i|} \quad (14)$$

(2) **Average Forgetting Rate** quantifying knowledge retention:

$$\text{FR} = \frac{1}{T-1}\sum_{i=1}^{T-1}(\text{Dice}_i^{\text{peak}} - \text{Dice}_i^{\text{final}}) \quad (15)$$

where $\text{Dice}_i^{\text{peak}}$ is the best validation performance on task $i$ observed immediately after training on that task, and $\text{Dice}_i^{\text{final}}$ is the performance after all tasks are learned. In Table 2, Params refers to trainable parameters, GPU denotes peak memory usage during training, and Time is relative to our method (1.0×).

**Implementation Details** We build upon the CLIPSeg architecture with a frozen backbone. LoRA modules (rank 8, $\alpha = 16$) are applied to the query, key, value, and output projections in both vision and text encoders; these adapters are trainable while the backbone parameters remain fixed. The text embeddings used for CRP modality assignment (Eq. 3) are extracted from the frozen CLIP text encoder before any LoRA adaptation, ensuring stable modality discovery throughout training. The CRP concentration parameter is set to $\alpha = 5.0$. For EWC, we use $\lambda = 5000$ with 200 samples for Fisher information estimation. Training uses AdamW with learning rate $1 \times 10^{-3}$ and weight decay $8 \times 10^{-5}$. Each task trains for up to 60 epochs with early stopping (patience 8, minimum 15 epochs). All images are resized to $352 \times 352$. Experiments are conducted on a single NVIDIA RTX 4090 GPU with batch size 16.

### 4.2. Comparison with State-of-the-Art Methods

**Baselines** We compare against four representative continual learning methods: EWC (Kirkpatrick et al., 2017), a regularization-based approach ($\lambda = 5000$); RAPF (Huang et al., 2024) and CL-LoRA (He et al., 2025), which employ parameter-efficient adapters with different fusion strategies; and MoE-Adapters (Yu et al., 2024), a mixture-of-experts approach ($K = 16$ experts with activate-freeze strategy). We also include *Sequential* fine-tuning without forgetting mitigation, and *Individual* models trained separately per task as an upper bound.

**Overall Results** Our method achieves the highest Dice score (73.3%) while maintaining the lowest forgetting rate (4.1%), significantly outperforming all baselines (Table 2). Compared to MoE-Adapters (Yu et al., 2024), which achieves 65.3% Dice with 51.9M parameters, our approach improves performance by 8.0% while using only 8.6M parameters—a 6× reduction. Although CL-LoRA (He et al., 2025) employs minimal parameters (0.05M), it suffers from higher forgetting (9.7%) and requires 1.5× more training time due to its knowledge distillation overhead. Classical regularization methods such as EWC (Kirkpatrick et al., 2017) and RAPF (Huang et al., 2024) show limited effectiveness in the medical imaging domain, achieving only 56.8% and 58.4% Dice respectively.

Figure 3 illustrates the performance dynamics across the 16-task sequence. Our method demonstrates remarkable stability: Cardiac US and Dermoscopy maintain near-constant

*Table 2.* Comparison of continual learning methods.

| Method | Dice (%) | Forget (%) | Params (M) | GPU (GB) | Time |
|---|---|---|---|---|---|
| Individual | 77.9 | – | 19.8 | 12.4 | – |
| Sequential | 48.0 ± 7.1 | 28.3 ± 7.7 | 1.2 | 5.8 | 0.7× |
| EWC | 56.8 ± 3.7 | 11.3 ± 3.5 | 1.2 | 5.8 | 0.9× |
| RAPF | 58.4 ± 1.7 | 7.2 ± 2.6 | 0.9 | 5.6 | 0.8× |
| CL-LoRA | 60.7 ± 2.0 | 9.7 ± 1.4 | 0.05 | 5.7 | 1.5× |
| MoE-Adapters | 65.3 ± 3.4 | 7.1 ± 3.2 | 51.9 | 13.3 | 1.2× |
| **Ours** | **73.3 ± 1.0** | **4.1 ± 0.8** | 8.6 | 12.4 | 1.0× |

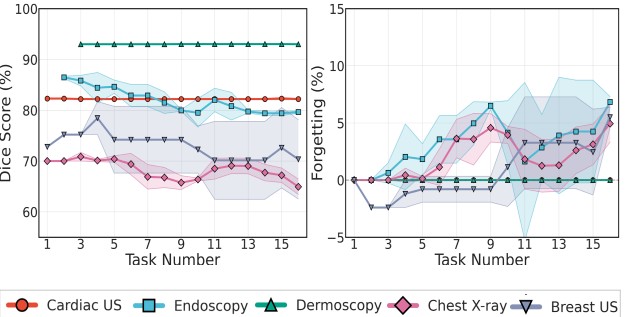

*Figure 3.* Performance analysis across all task orderings. Left: Dice score retention over the 16-task sequence. Right: Forgetting rate. Shaded regions indicate standard deviation across orderings. Negative forgetting indicates backward transfer.

Dice scores throughout training, while Endoscopy shows only modest degradation. Notably, the forgetting rates remain close to zero for most modalities, with occasional negative values indicating beneficial backward transfer. In contrast, Chest X-ray and Breast US exhibit higher variance, reflecting the inherent challenge of cross-modality continual learning. These results validate that our CRP-guided modality discovery effectively routes tasks to appropriate experts, preserving modality-specific knowledge while enabling positive knowledge transfer.

**Order Sensitivity Analysis** To evaluate the robustness to task ordering, we conduct experiments with four different task sequences: grouped (similar modalities consecutive), interleaved (alternating modalities), mixed (randomized), and reversed (inverse of grouped order). Full task sequences are provided in Appendix D. As shown in Figure 5, our method demonstrates consistent performance across all orderings, with Dice scores ranging from 0.72 to 0.74 and forgetting rates between 0.04 and 0.06. In contrast, MoE-Adapters exhibits greater sensitivity to task ordering, with Dice scores varying from 0.62 to 0.70 and higher forgetting rates (0.11-0.16). The narrow variability bands indicate effective adaptation to different task sequences, and the CRP-based semantic modality discovery mechanism does not require prior knowledge of optimal ordering.

*Table 3.* Comparison of different methods across datasets. Task order: CAMUS → Kvasir → ISIC → Airspace Op. → BUSI Ben. → ClinicDB → Atelectasis → ETIS → Cardiomeg. → CVC300 → BUSI Mal. → Edema → ColonDB → Enl. Cardio. → Pleural Eff. → Supp. Dev. Avg Dice shows segmentation performance; Avg FR (%) shows forgetting rate (lower is better). Bold indicates best performance among continual learning methods.

| Method | CAMUS | Kvasir | ISIC | Airspace Op. | BUSI Ben. | ClinicDB | Atelectasis | ETIS | Cardiomeg. | CVC300 | BUSI Mal. | Edema | ColonDB | Enl. Cardio. | Pleural Eff. | Supp. Dev. | Avg |
|---|---|---|---|---|---|---|---|---|---|---|---|---|---|---|---|---|---|
| *Dice (%) ↑* | | | | | | | | | | | | | | | | | |
| Individual | 83.4 | 86.6 | 93.1 | 72.1 | 83.6 | 84.4 | 67.8 | 70.8 | 73.3 | 92.3 | 72.6 | 77.4 | 76.5 | 72.3 | 66.7 | 73.1 | 77.9 |
| Sequential | 26.7 | 62.6 | 59.1 | 54.5 | 34.6 | 49.3 | 56.3 | 28.3 | 66.2 | 54.1 | 50.9 | 59.4 | 42.3 | 67.0 | 56.3 | 68.2 | 52.2 |
| EWC | 61.7 | 79.5 | 89.7 | 41.5 | 57.7 | 63.5 | 37.7 | 48.4 | 45.0 | 62.9 | 66.6 | 47.6 | 50.3 | 41.8 | 46.5 | 48.4 | 55.5 |
| RAPF | 17.3 | 75.8 | 87.0 | 68.8 | 57.9 | 58.4 | 65.4 | 40.1 | 68.9 | 49.1 | 69.3 | 70.9 | 44.4 | 66.7 | 54.0 | 72.0 | 60.4 |
| CL-LoRA | 37.4 | 66.4 | 84.6 | 68.6 | 53.3 | 56.5 | **66.8** | 30.7 | 70.9 | 61.1 | 69.4 | **74.6** | 50.3 | 68.8 | **66.7** | 72.1 | 62.4 |
| MoE-Adapters | 42.4 | 72.0 | 87.1 | **70.2** | **72.2** | 68.4 | 63.2 | 49.9 | **72.4** | 69.4 | **70.7** | 73.9 | 66.5 | 67.2 | 66.7 | 70.3 | 67.7 |
| **Ours** | **82.3** | **82.6** | **93.0** | 62.2 | 51.7 | **79.9** | 60.9 | **68.9** | 70.6 | **90.3** | 66.9 | 66.7 | **77.3** | **71.2** | 57.6 | **73.3** | **72.2** |
| *Forgetting (%) ↓* | | | | | | | | | | | | | | | | | |
| Sequential | 59.4 | 24.8 | 33.7 | 13.8 | 45.9 | 36.0 | 10.5 | 37.6 | 5.1 | 41.8 | 24.2 | 14.9 | 32.2 | 3.2 | 0.0 | 0.0 | 23.9 |
| EWC | 24.6 | 5.9 | 3.0 | 17.8 | 16.4 | 10.6 | 25.5 | 3.2 | 10.3 | 18.8 | 6.5 | 6.7 | 11.1 | 6.8 | 32.4 | 26.6 | 14.1 |
| RAPF | 42.4 | 4.4 | 4.6 | 0.5 | 20.9 | 4.8 | 1.4 | 7.6 | 3.8 | 26.7 | 0.7 | 3.6 | 10.5 | 1.0 | 8.2 | 0.0 | 8.8 |
| CL-LoRA | 6.6 | 13.9 | 6.5 | 0.8 | 25.5 | 17.1 | **0.0** | 22.3 | **0.0** | 13.7 | 2.3 | **0.0** | 14.5 | 0.1 | 0.1 | **0.0** | 8.4 |
| MoE-Adapters | 40.7 | 12.6 | 5.5 | **0.0** | **3.8** | 12.7 | 0.3 | 10.9 | 0.6 | 11.6 | **0.8** | 0.1 | 4.7 | 1.3 | **0.0** | 0.0 | 6.6 |
| **Ours** | **0.0** | **3.7** | **0.0** | 7.6 | 14.0 | **3.1** | 1.4 | **4.8** | 4.7 | **2.2** | **0.0** | 13.0 | **0.0** | **1.4** | 11.4 | **0.0** | **4.2** |

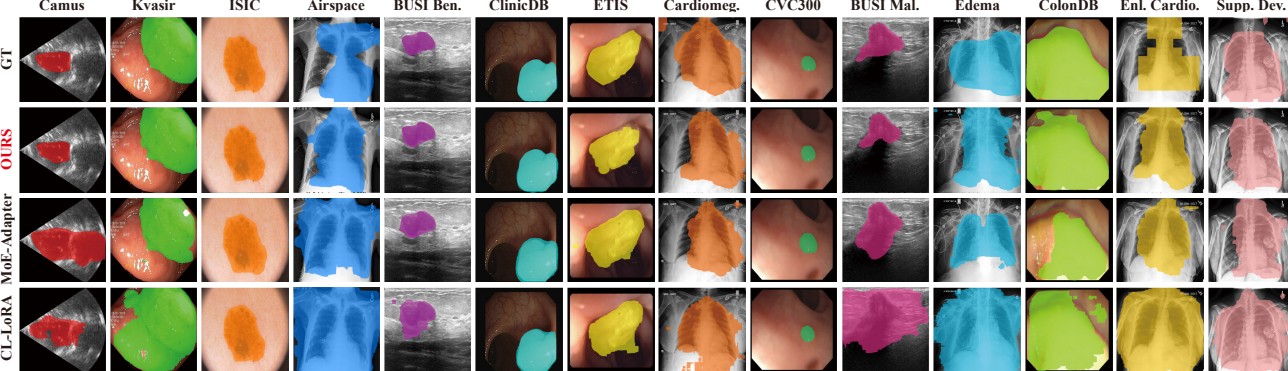

*Figure 4.* Visualization comparison of segmentation results cross representative tasks. Segmentation results from top row to bottom: Ground-truth, Ours, MoE-Adapters, and CL-LoRA.

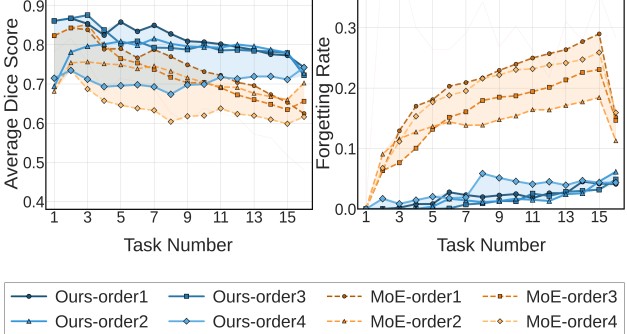

*Figure 5.* Order sensitivity analysis evaluating robustness to task ordering across four sequences. Our method (blue) maintains stable performance with narrow variability bands, while MoE-Adapters (orange) exhibits larger fluctuations.

**Comparison on Interleaved Task Order** Table 3 presents detailed per-task results on the mixed 16-task sequence. Our method achieves the best average Dice (72.2%) and lowest forgetting rate (4.2%), with strong performance on early tasks such as CAMUS (82.3%) and CVC300 (90.3%). This indicates effective knowledge retention throughout the learning process. While MoE-Adapters (Yu et al., 2024) and CL-LoRA (He et al., 2025) achieve competitive results on certain Chest X-ray tasks, they suffer from severe forgetting on others. Results on additional task orders are provided in Appendix F. Figure 4 provides qualitative comparisons: our method produces segmentation masks closest to ground truth, with sharper boundaries and fewer false positives, whereas baselines often exhibit incomplete regions or miss small structures.

### 4.3. Ablation Study

**Module Ablation Analysis**  Table 4 evaluates each component's contribution. Removing EWC increases forgetting from 4.09% to 5.41%, confirming that intra-modality regularization (Kirkpatrick et al., 2017) preserves knowledge within modality clusters. The most significant degradation occurs without CRP: forgetting rises to 15.55% as incompatible modalities interfere in a shared adapter. The Single LoRA baseline exhibits catastrophic forgetting (27.34%), consistent with naive fine-tuning. Removing LoRA yields near-zero forgetting but poor Dice (45.39%), as the frozen backbone cannot adapt to new tasks. These results confirm that CRP provides modality isolation, LoRA enables adaptation, and EWC consolidates intra-modality knowledge.

*Table 4.* Ablation study of different components.

| Config. | CRP | LoRA | EWC | Dice | Forgetting |
|---|---|---|---|---|---|
| Full Model | √ | √ | √ | **73.33** | **4.09** |
| w/o EWC | √ | √ | × | 71.92 | 5.41 |
| w/o CRP | × | √ | √ | 57.59 | 15.55 |
| Single LoRA | × | √ | × | 46.94 | 27.34 |
| w/o LoRA | √ | × | √ | 45.39 | 0.03 |

**Loss Function Study**  Table 5 ablates our loss components. The full model ($\mathcal{L}_{Seg} + \mathcal{L}_{Dice} + \mathcal{L}_{EWC}$) achieves the best Dice (73.33%) with consistent modality discovery (5 clusters). Removing $\mathcal{L}_{EWC}$ increases forgetting from 4.09% to 5.25% and causes unstable modality allocation (5–6 clusters) as feature drift leads CRP to create redundant experts. Removing $\mathcal{L}_{Dice}$ achieves the lowest forgetting (3.66%) but sacrifices segmentation accuracy (71.87%), reflecting a stability-plasticity trade-off. Using only $\mathcal{L}_{Seg}$ yields the lowest performance (70.27%) with highly variable clustering (4–6 modalities). These results confirm that $\mathcal{L}_{EWC}$ stabilizes features for consistent CRP routing, while $\mathcal{L}_{Dice}$ directly optimizes segmentation quality.

*Table 5.* Ablation study of loss components.

| Config. | Dice | Forgetting | #Mod. |
|---|---|---|---|
| $\mathcal{L}_{Seg}$ | 70.27 | 4.97 | 4–6 |
| $\mathcal{L}_{Seg} + \mathcal{L}_{EWC}$ | 71.87 | **3.66** | 5 |
| $\mathcal{L}_{Seg} + \mathcal{L}_{Dice}$ | 72.41 | 5.25 | 5–6 |
| $\mathcal{L}_{Seg} + \mathcal{L}_{Dice} + \mathcal{L}_{EWC}$ | **73.33** | 4.09 | 5 |

### 4.4. Semantic Modality Discovery Analysis

**Visualization**  To understand how our method organizes knowledge across tasks, we apply t-SNE to visualize the learned prompt embeddings (Figure 6). The visualization reveals that CRP automatically discovers five distinct semantic modality clusters, grouping tasks by their underlying imaging characteristics rather than their arrival order. No-

tably, CRP separates cardiac ultrasound (CAMUS) from breast ultrasound (BUSI), recognizing that these tasks require different feature representations despite sharing the same physical imaging modality. This emergent structure enables our method to route incoming tasks to appropriate experts, facilitating forward transfer within semantic modalities while isolating unrelated domains.

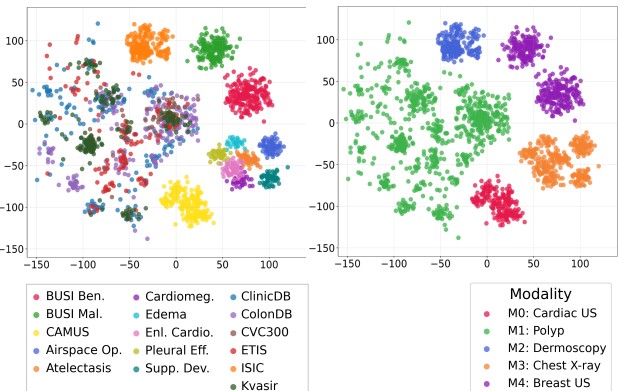

*Figure 6.* t-SNE visualization of prompt embeddings. Left: colored by dataset (16 classes). Right: colored by semantic modality discovered via CRP (5 clusters). Tasks with similar clinical semantics are automatically grouped together.

**Comparison with Physical Modality Grouping**  Table 6 compares modality assignment strategies for continual learning. Grouping tasks by physical imaging modality, namely the acquisition device type (Ultrasound, Endoscopy, Dermoscopy, X-ray; K=4), merges cardiac and breast ultrasound into a single modality since both are acquired via ultrasound despite involving fundamentally different anatomical structures. This causes parameter interference and elevated forgetting (9.23%). Our CRP-based approach automatically discovers K=5 semantic modalities from clinical text prompts, correctly separating cardiac from breast ultrasound based on their distinct anatomical contexts, improving Dice by 7.6% and reducing forgetting by 5.1%. We further validate the necessity of this separation by merging all 10 cross-modality adapter pairs via Fisher-weighted averaging. All result in degradation (Appendix H).

*Table 6.* Impact of modality discovery on continual learning performance.

| Modality Assignment | K | Dice | Forgetting |
|---|---|---|---|
| Physical imaging type | 4 | 65.75 | 9.23 |
| CRP discovered (Ours) | 5 | **73.33** | **4.09** |

**Text-Based vs. Visual-Based Clustering**  We compare text-only and visual-only clustering using embeddings from the same CLIP backbone. Table 7 both similarity statistics and discovered $K$ across $\alpha$ values. Text embeddings pro-

vide a substantially larger intra-/inter-group gap ($\sim$0.50 vs. $\sim$0.22 for visual), with the critical case being cardiac vs. breast ultrasound: visual similarity 0.95+ but text similarity $\sim$0.45. Consequently, text-only clustering discovers stable $K$=5 across all $\alpha$, while visual-only and dual-track produce inconsistent $K$ that shifts with $\alpha$. Adding visual features degrades the correct $K$=5 structure that text-only achieves.

*Table 7.* Clustering signal comparison.

| Embedding | Intra | Inter | Gap | Discovered $K$ |
|---|---|---|---|---|
| Text-only | 0.95+ | 0.45 | $\sim$0.50 | 5 |
| Visual-only | 0.95+ | 0.73 | $\sim$0.22 | 1–4 |

**Robustness to Encoder Choice**   To verify that the discovered structure is not an artifact of the specific text encoder, we replace the CLIP encoder with 9 alternatives spanning 4 training paradigms, including medical contrastive models (Eslami et al., 2023; Zhang et al., 2023; luhuitong, 2024), general contrastive models (Cherti et al., 2023), and non-CLIP architectures combining contrastive learning with generative, masked modeling, or matching objectives (Singh et al., 2022; Li et al., 2022; Yu et al., 2022). As shown in Table 8, all 10 encoders discover identical $K$=5 with the same cluster membership at $\alpha$=5, confirming that the semantic modality structure is an intrinsic property of the data. Encoders differ only in embedding space compactness, which shifts the $\alpha$ range producing $K$=5: our default CLIP encoder maintains $K$=5 across the widest range.

We also tested two non-contrastive encoders, SigLIP and S-PubMedBERT, both of which produced $K$=1 across all $\alpha$ values due to insufficient inter-task separation in their embedding spaces. Our method requires text encoders trained with contrastive objectives.

*Table 8.* Encoder sensitivity analysis. All 10 encoders discover identical $K$=5 with the same cluster membership at $\alpha$=5.

| Encoder | $\alpha$=2 | $\alpha$=5 | $\alpha$=7 | $\alpha$=10 |
|---|---|---|---|---|
| *Baseline* | | | | |
| CLIP, CLIPSeg built-in (Ours) | 5 | 5 | 5 | 5 |
| *Medical VLP (contrastive, medical)* | | | | |
| PubMedCLIP | 5 | 5 | 5 | 5 |
| BiomedCLIP | 1 | 2 | 5 | 5 |
| MedICaT-ROCO | 5 | 5 | 5 | 10 |
| *General VLP (contrastive, LAION)* | | | | |
| OpenCLIP-B/32 | 2 | 5 | 5 | 5 |
| OpenCLIP-B/16 | 2 | 5 | 5 | 5 |
| OpenCLIP-L/14 | 4 | 5 | 5 | 5 |
| *Non-CLIP (diverse training paradigms)* | | | | |
| FLAVA | 4 | 5 | 5 | 5 |
| BLIP | 3 | 5 | 5 | 5 |
| CoCa | 2 | 5 | 5 | 5 |

**Prompt Robustness**   We evaluate prompt robustness at two levels. CRP discovers identical $K$=5 under both detailed prompts (e.g., "Left ventricular cavity of oval shape in four-chamber view...") and concise prompts (e.g., "round polyp"), achieving 100% clustering consistency. Full prompt templates are in Appendix C. Table 9 evaluates robustness under clinically realistic perturbations including abbreviations, typos, keyword drops, and word reordering. All plausible noise conditions preserve $K$=5. Degradation occurs only under extreme perturbations ($>$30% typos or $>$50% keyword drop), where CRP defaults to increased parameter sharing rather than creating spurious clusters.

*Table 9.* CRP robustness to prompt perturbations. Clinically realistic perturbations (above the line) all preserve $K$=5.

| Perturbation | Level | $K$ |
|---|---|---|
| None (original) | – | 5 |
| Clinical abbreviation | – | 5 |
| Realistic typo | 10–20% | 5 |
| Keyword drop | 20–30% | 5 |
| Word shuffle | – | 5 |
| Realistic typo | 30% | 3 |
| Keyword drop | 50% | 1 |
| Generic prompt | – | 1 |

## 5. Conclusion

We presented MedCRP-CL, a framework for continual medical image segmentation that automatically discovers semantic modality structure from clinical text prompts via the Chinese Restaurant Process. Combined with modality-specific LoRA adapters and intra-modality EWC regularization, our method achieves cross-modality isolation while enabling within-modality knowledge transfer. The framework is also replay-free, requiring no storage of raw patient data. Experiments demonstrate state-of-the-art performance with significantly reduced forgetting and fewer parameters. These results suggest that leveraging semantic information from clinical text prompts provides a principled approach to structure discovery in continual learning.

## Impact Statement

This paper presents work whose goal is to advance continual learning for medical image segmentation. Our method is designed with privacy preservation as a core principle: by storing only aggregate statistics rather than raw patient data, it aligns with healthcare privacy regulations such as HIPAA and GDPR, potentially enabling deployment in privacy-sensitive clinical environments.

While improved medical image analysis could benefit clinical diagnosis and treatment planning, deployment in real clinical settings would require appropriate prospective vali-

dation and regulatory approval. We encourage practitioners to use such AI-assisted tools as decision support rather than replacements for clinical judgment. We do not foresee any immediate negative societal consequences specific to our methodological contributions beyond those common to the field of medical AI.

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

# A. Theoretical Analysis

We provide theoretical guarantees for our approach.

**Proposition A.1** (Modality Clustering Consistency). *Assume intra-modality similarities follow $\mathcal{N}(\mu_{intra}, \sigma_{intra}^2)$ and inter-modality similarities follow $\mathcal{N}(\mu_{inter}, \sigma_{inter}^2)$ with separation $\Delta = \mu_{intra} - \mu_{inter} > 0$. If $\Delta > 2(\sigma_{intra} + \sigma_{inter})$, then MAP modality assignment achieves zero clustering error as $t \to \infty$ with probability 1.*

*Proof.* The log-likelihood ratio $\ell(s)$ defined in Eq. 5 acts as a binary classifier between same-modality and different-modality hypotheses. Under the Gaussian assumption, the optimal decision boundary is at:

$$s^* = \frac{\mu_{\text{intra}}\sigma_{\text{inter}}^2 + \mu_{\text{inter}}\sigma_{\text{intra}}^2}{\sigma_{\text{intra}}^2 + \sigma_{\text{inter}}^2} \tag{16}$$

The classification error is bounded by:

$$P_{\text{error}} = P(s_{\text{intra}} < s^*) + P(s_{\text{inter}} > s^*) \leq 2\exp\left(-\frac{\Delta^2}{8(\sigma_{\text{intra}}^2 + \sigma_{\text{inter}}^2)}\right) \tag{17}$$

by the Chernoff bound. The separation condition $\Delta > 2(\sigma_{\text{intra}} + \sigma_{\text{inter}})$ ensures $P_{\text{error}} \to 0$ exponentially.

By the strong law of large numbers, Welford's online estimates converge almost surely: $\hat{\mu}_{\text{intra}} \xrightarrow{a.s.} \mu_{\text{intra}}$ and $\hat{\sigma}_{\text{intra}} \xrightarrow{a.s.} \sigma_{\text{intra}}$ (similarly for inter-modality statistics). Combined with the CRP's exchangeability property and de Finetti's theorem, the posterior concentrates on the true partition as $t \to \infty$. $\qquad\square$

**Proposition A.2** (Modality-Isolated Forgetting Bound). *Let $\mathcal{L}_t^*$ be the optimal loss achievable on task $t$. Under modality-isolated EWC with coefficient $\lambda$, the forgetting after training $m$ subsequent tasks within the same modality satisfies:*

$$\mathcal{L}_t(\theta_k^{(m)}) - \mathcal{L}_t^* \leq \frac{1}{\lambda \cdot \lambda_{\min}(\bar{F}_k)} \sum_{j=1}^{m} \|\nabla_{\theta_k}\mathcal{L}_{t+j}\|^2 \tag{18}$$

*where $\lambda_{\min}(\bar{F}_k)$ is the minimum eigenvalue of the consolidated Fisher matrix.*

*Proof.* Following the analysis of Kirkpatrick et al. (2017), the EWC penalty constrains parameter movement in directions important for previous tasks. For task $t$ in modality $k$, let $\theta_k^{(0)}$ be the parameters after training task $t$ (which achieves $\mathcal{L}_t^*$), and $\theta_k^{(m)}$ be parameters after training $m$ subsequent tasks.

The EWC objective ensures:

$$\|\theta_k^{(m)} - \theta_k^{(0)}\|_{\bar{F}_k}^2 \leq \frac{1}{\lambda}\sum_{j=1}^{m}\mathcal{L}_{t+j}(\theta_k^{(j-1)}) \tag{19}$$

Since $\theta_k^{(0)}$ minimizes $\mathcal{L}_t$, we have $\mathcal{L}_t(\theta_k^{(0)}) = \mathcal{L}_t^*$ and $\nabla\mathcal{L}_t(\theta_k^{(0)}) = 0$. By Taylor expansion:

$$\mathcal{L}_t(\theta_k^{(m)}) - \mathcal{L}_t^* \approx \frac{1}{2}(\theta_k^{(m)} - \theta_k^{(0)})^\top H_t(\theta_k^{(m)} - \theta_k^{(0)}) \tag{20}$$

Using $H_t \preceq \bar{F}_k$ (Fisher information approximates Hessian at convergence) and $\|x\|_{\bar{F}_k}^2 \geq \lambda_{\min}(\bar{F}_k)\|x\|^2$:

$$\mathcal{L}_t(\theta_k^{(m)}) - \mathcal{L}_t^* \leq \frac{1}{2\lambda_{\min}(\bar{F}_k)}\|\theta_k^{(m)} - \theta_k^{(0)}\|_{\bar{F}_k}^2 \leq \frac{1}{\lambda \cdot \lambda_{\min}(\bar{F}_k)}\sum_{j=1}^{m}\|\nabla_{\theta_k}\mathcal{L}_{t+j}\|^2 \tag{21}$$

Critically, this bound involves only tasks within modality $k$. Tasks in other modalities use separate parameters and do not contribute to forgetting on task $t$. $\qquad\square$

**Implications.** Proposition A.1 guarantees that our modality discovery mechanism correctly identifies the underlying structure given sufficient separation in prompt embeddings. In our experiments, we observe $\mu_{\text{intra}} \approx 0.94$ and $\mu_{\text{inter}} \approx 0.51$ with $\sigma_{\text{intra}} \approx 0.05$ and $\sigma_{\text{inter}} \approx 0.10$, yielding $\Delta \approx 0.43$ which substantially exceeds the required separation of $2(\sigma_{\text{intra}} + \sigma_{\text{inter}}) = 0.30$. Proposition A.2 shows that forgetting scales only with tasks within a modality, not the total task count—a key advantage over global EWC approaches.

## B. Datasets Description

**Kvasir-SEG (2020)** (Jha et al., 2020): Endoscopic polyp segmentation dataset containing 1000 colonoscopy images with pixel-level annotations. Images feature diverse polyp morphologies, sizes, and locations within the colon, representing real clinical scenarios for computer-aided polyp detection during routine colonoscopy procedures.

**ClinicDB (2015)** (Bernal et al., 2015): Clinical endoscopic database for polyp segmentation comprising 612 colonoscopy frames with corresponding ground truth masks. Dataset emphasizes challenging cases with varying illumination conditions, polyp textures, and anatomical backgrounds commonly encountered in clinical practice.

**ETIS (2014)** (Silva et al., 2014): Endoscopic polyp segmentation dataset with 196 high-resolution colonoscopy images. Dataset emphasizes challenging polyp detection scenarios including flat lesions, small polyps, and cases with poor visibility conditions that test segmentation algorithm robustness in clinical environments.

**ColonDB (2016)** (Tajbakhsh et al., 2016): Comprehensive colonoscopy database containing 380 images with polyp segmentation masks. Dataset includes diverse polyp types, anatomical locations, and imaging conditions, providing extensive coverage of endoscopic appearance variations encountered during routine colonoscopic examinations.

**CVC300 (2017)** (Vázquez et al., 2017): Colonoscopy video database comprising 60 polyp segmentation cases extracted from endoscopic sequences. Dataset focuses on temporal consistency and motion artifacts in video-based polyp detection, essential for real-time clinical applications during live colonoscopy procedures.

**ISIC 2016 (2016)** (Codella et al., 2018): International Skin Imaging Collaboration dataset containing 1279 dermoscopy images for melanoma segmentation. Dataset includes diverse skin lesion types, pigmentation patterns, and imaging artifacts, representing global dermatological imaging standards for automated skin cancer detection.

**CAMUS (2019)** (Leclerc et al., 2019): Cardiac ultrasound segmentation dataset containing 6000 echocardiographic images with left ventricle annotations. Dataset covers multiple cardiac views and pathological conditions, enabling comprehensive evaluation of automated cardiac function assessment in clinical echocardiography.

**BUSI (2020)** (Al-Dhabyani et al., 2020): Breast ultrasound segmentation dataset comprising 780 images with lesion annotations. Dataset includes benign and malignant breast masses with varying echogenicity patterns, supporting development of automated breast cancer screening and diagnostic assistance tools. We split this dataset by pathology into BUSI-Benign and BUSI-Malignant subsets to enable fine-grained evaluation across lesion types.

**CheXlocalize (2019)** (Irvin et al., 2019): Chest X-ray pathology localization dataset containing 2177 radiographs with bounding box annotations. Dataset covers multiple thoracic pathologies including pneumonia, effusions, and nodules, enabling evaluation of automated radiological interpretation and diagnosis assistance systems. We create seven pathology-specific subsets (Airspace Opacity, Atelectasis, Cardiomegaly, Edema, Enlarged Cardiomediastinum, Pleural Effusion, and Support Devices) to evaluate segmentation performance on individual clinical findings.

# C. Prompt Templates

*Table 10.* Prompt templates for all 16 datasets, organized by discovered semantic modality.

| Dataset | Concise | Detailed |
|---------|---------|----------|
| *Semantic Modality 0: Cardiac Ultrasound* | | |
| CAMUS | Left ventricular cavity in the cardiac ultrasound. | Left ventricular cavity of oval shape in four-chamber view in the cardiac ultrasound at end of the diastole cycle of a 18-year-old male with good image quality. |
| *Semantic Modality 1: Endoscopy (Polyp Segmentation)* | | |
| Kvasir-SEG | round polyp | One medium pink round polyp which is a projecting growth of tissue located in top left of the image. |
| ClinicDB | oval polyp | Two large white oval polyp which is a projecting growth of tissue located in center, bottom right of the image. |
| CVC-300 | triangular polyp | One small brown triangular polyp which is a projecting growth of tissue located in bottom right of the image. |
| ColonDB | kidney polyp | One large orange kidney polyp which is a projecting growth of tissue located in center of the image. |
| ETIS | circle polyp | Two small yellow circle polyp which is a projecting growth of tissue located in top, right of the image. |
| *Semantic Modality 2: Dermoscopy (Skin Lesion Segmentation)* | | |
| ISIC | circle skin melanoma | One large blue circle skin melanoma which is a dark sore with irregular texture located in center of the image. |
| *Semantic Modality 3: Chest X-ray* | | |
| CheX-Airspace Opacity | Airspace Opacity in a chest Xray. | Airspace Opacity in a Chest Xray. Enlarged Cardiomediastinum, Cardiomegaly, Lung Opacity are present. |
| CheX-Atelectasis | Atelectasis in a chest Xray. | Atelectasis in a Chest Xray. Enlarged Cardiomediastinum, Lung Opacity, Atelectasis, Pleural Effusion are present. |
| CheX-Cardiomegaly | Cardiomegaly in a chest Xray. | Cardiomegaly in a Chest Xray. Enlarged Cardiomediastinum, Cardiomegaly, Lung Opacity, Edema are present. |
| CheX-Edema | Edema in a chest Xray. | Edema in a Chest Xray. Lung Opacity, Edema, Consolidation, Atelectasis are present. |
| CheX-Enlarged Card. | Enlarged Cardiomediastinum in a chest Xray. | Enlarged Cardiomediastinum in a Chest Xray. Enlarged Cardiomediastinum, Cardiomegaly, Lung Opacity are present. |
| CheX-Pleural Effusion | Pleural Effusion in a chest Xray. | Pleural Effusion in a Chest Xray. Lung Opacity, Atelectasis, Pleural Effusion, Support Devices are present. |
| CheX-Support Devices | Support Devices in a chest Xray. | Support Devices in a Chest Xray. Enlarged Cardiomediastinum, Cardiomegaly, Support Devices are present. |
| *Semantic Modality 4: Breast Ultrasound* | | |
| BUSI-Benign | Benign tumor in the breast ultrasound image. | Five small, medium circle-shaped benign tumors at the top right, top, center, right, right in the breast ultrasound image. |
| BUSI-Malignant | Malignant tumor in the breast ultrasound image. | One large irregular-shaped malignant tumor at the center in the breast ultrasound image. |

# D. Task Orders

To evaluate robustness to task arrival sequences, we test four distinct orderings spanning 16 segmentation tasks. Table 11 shows all orderings used in our experiments.

*Table 11.* Task orderings used in experiments. Each order presents 16 medical image segmentation tasks in a different sequence.

| Order | Task Sequence |
| --- | --- |
| grouped | CAMUS → Kvasir → ClinicDB → ETIS → CVC-300 → ColonDB → ISIC → CheX-Airspace → CheX-Atelectasis → CheX-Cardiomegaly → CheX-Edema → CheX-Enlarged → CheX-Pleural → CheX-Support → BUSI-Benign → BUSI-Malignant |
| *Rationale* | *Tasks grouped by semantic modality, simulating sequential adoption of imaging equipment in a clinical setting.* |
| reversed | BUSI-Malignant → BUSI-Benign → CheX-Support → CheX-Pleural → CheX-Enlarged → CheX-Edema → CheX-Cardiomegaly → CheX-Atelectasis → CheX-Airspace → ISIC → ColonDB → CVC-300 → ETIS → ClinicDB → Kvasir → CAMUS |
| *Rationale* | *Reverse of grouped order to evaluate sensitivity to arrival direction while preserving modality clustering.* |
| interleaved | CAMUS → Kvasir → ISIC → CheX-Airspace → BUSI-Benign → ClinicDB → CheX-Atelectasis → ETIS → CheX-Cardiomegaly → CVC-300 → BUSI-Malignant → CheX-Edema → ColonDB → CheX-Enlarged → CheX-Pleural → CheX-Support |
| *Rationale* | *Alternating modalities to simulate realistic clinical deployment where new imaging types arrive unpredictably.* |
| mixed | CheX-Airspace → Kvasir → CAMUS → BUSI-Benign → ClinicDB → CheX-Cardiomegaly → ISIC → CheX-Atelectasis → ETIS → BUSI-Malignant → CheX-Edema → CVC-300 → ColonDB → CheX-Pleural → CheX-Enlarged → CheX-Support |
| *Rationale* | *Randomized task arrival to stress-test modality discovery under arbitrary task sequences.* |

# E. CRP Assignment Reliability

## E.1. Cold-Start Assignment Trace

Before Gaussian activation, the CRP relies on a parameter-free logit-based likelihood with 0.5 as the decision boundary. Table 12 shows the first four task assignments under four different orderings. Same-modality pairs consistently produce similarity >0.79 while cross-modality pairs produce similarity <0.57, providing a margin of 0.22+ for reliable separation. All pre-Gaussian assignments are consistent with the final converged $K$=5 structure across all orderings.

*Table 12.* Per-task assignment trace during cold-start (first 4 tasks) under 4 orderings. Values indicate similarity logits; JOIN/NEW denotes the CRP decision.

| Order | Task 1 | Task 2 | Task 3 | Task 4 |
|---|---|---|---|---|
| Grouped | Kvasir NEW (0.474) | ClinicDB JOIN-M1 (0.991) | ETIS JOIN-M1 (0.980) | CVC300 JOIN-M1 (0.992) |
| Interleaved | Kvasir NEW (0.474) | ISIC NEW (0.548) | ChexOp. NEW (0.488) | BUSI-B. NEW (0.568) |
| Mixed | Kvasir NEW (0.305) | CAMUS NEW (0.488) | BUSI-B. NEW (0.568) | ClinicDB JOIN-M1 (0.991) |
| Reversed | BUSI-B. JOIN-M0 (0.915) | ChexSup. NEW (0.493) | ChexEnl. JOIN-M1 (0.790) | ChexEdema JOIN-M1 (0.870) |

## E.2. Gaussian Estimate Stabilization

Table 13 tracks the intra- and inter-modality Gaussian statistics as observations accumulate. The intra/inter gap remains above 0.45 throughout and stabilizes within 2–3 observations after activation. Final distributions are consistent across all 4 task orderings (intra_mean: 0.921–0.931, inter_mean: 0.429–0.473).

*Table 13.* Gaussian estimate convergence. The intra/inter similarity gap stabilizes within 2–3 observations.

| Intra $n$ | $\mu_{\text{intra}}$ | $\sigma_{\text{intra}}$ | $\mu_{\text{inter}}$ | Gap |
|---|---|---|---|---|
| 1 | 0.980 | 0.100 | 0.445 | 0.535 |
| 2 | 0.925 | 0.078 | 0.448 | 0.477 |
| 3 | 0.947 | 0.067 | 0.444 | 0.503 |
| 8 | 0.921 | 0.064 | 0.429 | 0.492 |

# F. Results Across Different Task Orderings

*Table 14.* MedCRP-CL performance on `grouped` order: CAMUS → Kvasir → ClinicDB → ETIS → CVC300 → ColonDB → ISIC → Airspace Op. → Atelectasis → Cardiomeg. → Edema → Enl. Cardio. → Pleural Eff. → Supp. Dev. → BUSI Ben. → BUSI Mal.

| | CAMUS | Kvasir | ClinicDB | ETIS | CVC300 | ColonDB | ISIC | Airspace Op. | Atelectasis | Cardiomeg. | Edema | Enl. Cardio. | Pleural Eff. | Supp. Dev. | BUSI Ben. | BUSI Mal. | Avg |
|---|---|---|---|---|---|---|---|---|---|---|---|---|---|---|---|---|---|
| Dice (%) ↑ | 82.3 | 82.6 | 79.9 | 68.9 | 90.3 | 77.3 | 93.0 | 62.3 | 60.8 | 69.6 | 66.7 | 71.0 | 57.7 | 73.8 | 76.8 | 74.7 | 74.2 |
| Fgt. (%) ↓ | 0.0 | 3.7 | 3.1 | 4.8 | 2.2 | 0.0 | 0.0 | 7.4 | 0.5 | 5.4 | 12.9 | 1.5 | 9.4 | 0.0 | 3.3 | 0.0 | 3.4 |

*Table 15.* MedCRP-CL performance on `mixed` order: Airspace Op. → Kvasir → CAMUS → BUSI Ben. → ClinicDB → Cardiomeg. → ISIC → Atelectasis → ETIS → BUSI Mal. → Edema → CVC300 → ColonDB → Pleural Eff. → Enl. Cardio. → Supp. Dev.

| | Airspace Op. | Kvasir | CAMUS | BUSI Ben. | ClinicDB | Cardiomeg. | ISIC | Atelectasis | ETIS | BUSI Mal. | Edema | CVC300 | ColonDB | Pleural Eff. | Enl. Cardio. | Supp. Dev. | Avg |
|---|---|---|---|---|---|---|---|---|---|---|---|---|---|---|---|---|---|
| Dice (%) ↑ | 57.8 | 82.6 | 82.0 | 76.6 | 78.8 | 68.3 | 93.0 | 56.4 | 67.9 | 75.3 | 62.9 | 90.0 | 77.3 | 51.9 | 69.4 | 73.2 | 72.7 |
| Fgt. (%) ↓ | 12.2 | 4.2 | 0.0 | 5.1 | 5.0 | 5.9 | 0.0 | 5.1 | 4.9 | 0.0 | 17.0 | 2.1 | 0.0 | 16.4 | 4.0 | 0.0 | 5.1 |

*Table 16.* MedCRP-CL performance on `reversed` order: BUSI Mal. → BUSI Ben. → Supp. Dev. → Enl. Cardio. → Edema → Cardiomeg. → Atelectasis → Airspace Op. → ISIC → ColonDB → CVC300 → ETIS → ClinicDB → Kvasir → Pleural Eff. → CAMUS.

| | BUSI Mal. | BUSI Ben. | Supp. Dev. | Enl. Cardio. | Edema | Cardiomeg. | Atelectasis | Airspace Op. | ISIC | ColonDB | CVC300 | ETIS | ClinicDB | Kvasir | Pleural Eff. | CAMUS | Avg |
|---|---|---|---|---|---|---|---|---|---|---|---|---|---|---|---|---|---|
| Dice (%) ↑ | 71.1 | 79.2 | 65.0 | 63.7 | 73.0 | 65.9 | 56.2 | 69.4 | 93.1 | 72.9 | 87.6 | 66.2 | 80.2 | 86.8 | 65.0 | 82.6 | 73.6 |
| Fgt. (%) ↓ | 1.7 | 0.0 | 6.7 | 7.6 | 4.5 | 9.4 | 10.9 | 2.4 | 0.0 | 4.0 | 6.1 | 3.5 | 3.4 | 0.0 | 3.0 | 0.0 | 3.9 |

## G. Prompt Perturbation Examples

Table 17 provides the full noisy prompt examples used in the prompt robustness evaluation (Table 9). All perturbations are applied to the same original prompt describing a cardiac ultrasound task: "Left ventricular cavity of rectangle shape in two-chamber view of the heart at end of the diastole cycle of a 56-year-old f with good image quality."

*Table 17.* Prompt perturbation examples corresponding to Table 9.

| Perturbation | Noisy Example |
|---|---|
| *Clinically Realistic* | |
| None (original) | Left ventricular cavity of rectangle shape in two-chamber view of the heart at end of the diastole cycle of a 56-year-old f with good image quality. |
| Abbreviation | Left ventricular cavity of rectangle shape in 2CV of the heart at ED of a 56-year-old f with good IQ. |
| Typo 10% | Left ventricular cavity of rectangle shape in two-chamber veiw of the heart at end of the diastole cycle of a 56-year-old f with good image qualty. |
| Typo 20% | Left venricular cavit of rectangle sshape in two-chambr vie of the heart at end of the diastole cycle of a 56-year-old f with good image quality. |
| Drop 20% | Left ventricular cavity of rectangle shape in two-chamber view the heart end of the cycle of 56-year-old f with good image. |
| Drop 30% | Left ventricular cavity of rectangle two-chamber view the heart end of the cycle of 56-year-old f with good image. |
| Shuffle | diastole at of quality. a in shape the the with heart of ventricular of good end cavity cycle image rectangle f two-chamber view Left of 56-year-old. |
| *Extreme Perturbations* | |
| Typo 30% | Left venricular cavit of rectangle sshape in two-chambr vie of tthe heart at ennd of tge diastle cycle of a 56-year-old f with good image quality. |
| Drop 50% | Left cavity of rectangle two-chamber view end of cycle 56-year-old f good image. |
| Generic | segment the region. |

## H. Fisher-Weighted Merge Analysis

We merge each pair of modality-specific LoRA adapters using Fisher-weighted averaging, with Fisher information matrices already computed during EWC. After merging, all affected tasks are re-adapted for 5 epochs until convergence. Results are shown in Table 18. All 10 pairs degrade, confirming that each discovered semantic modality captures distinct representations that cannot be consolidated without loss.

*Table 18.* Fisher-weighted merge results across all 10 cross-modality pairs. All merges degrade performance despite re-adaptation.

| Merge Pair | Modalities | Before | After | $\Delta$ |
|---|---|---|---|---|
| M0+M1 | Cardiac+Polyp | 0.793 | 0.398 | $-0.395$ |
| M0+M2 | Cardiac+Dermoscopy | 0.871 | 0.643 | $-0.228$ |
| M0+M3 | Cardiac+ChestXR | 0.686 | 0.538 | $-0.148$ |
| M0+M4 | Cardiac+BreastUS | 0.772 | 0.457 | $-0.315$ |
| M1+M2 | Polyp+Dermoscopy | 0.812 | 0.654 | $-0.158$ |
| M1+M3 | Polyp+ChestXR | 0.718 | 0.544 | $-0.174$ |
| M1+M4 | Polyp+BreastUS | 0.778 | 0.546 | $-0.232$ |
| M2+M3 | Dermoscopy+ChestXR | 0.701 | 0.685 | $-0.016$ |
| M2+M4 | Dermoscopy+BreastUS | 0.812 | 0.754 | $-0.058$ |
| M3+M4 | ChestXR+BreastUS | 0.687 | 0.647 | $-0.039$ |

