# OpenReview forum: "MedCRP-CL: Continual Medical Image Segmentation via Bayesian Nonparametric Semantic Modality Discovery"
_ICML.cc/2026/Conference — ICML 2026 regular_

### Official Review · Reviewer_KSx4 · 2026-03-09

**Soundness:** 3
**Presentation:** 3
**Significance:** 3
**Originality:** 4
**Overall Recommendation:** 5
**Confidence:** 4

**Summary:**

### Review
This paper addresses an important problem in continual medical image segmentation and proposes a neat structure-aware solution based on Bayesian nonparametric semantic modality discovery. The idea of using prompt semantics to dynamically group tasks and activate modality-specific adapters is interesting and well aligned with the heterogeneous nature of medical data. Overall, the method is technically sound, empirically strong, and appears to offer a practical replay-free framework for medical continual learning.

**Compliance With Llm Reviewing Policy:**

Affirmed.

**Final Justification:**

I find the work to have some value and practical relevance. The rebuttal improves my confidence slightly, and I accordingly raise my evaluation.

**Key Questions For Authors:**

The paper attributes its gains to semantic modality discovery, but the current formulation appears to cluster tasks primarily based on prompt embedding similarity rather than jointly modeling visual/task-level relatedness. Could the authors clarify whether the discovered groups should be interpreted as true latent visual modalities, or more as text-guided proxies for task affinity?

**Limitations:**

Yes.

**Strengths And Weaknesses:**

### Pros
The paper is well motivated, and the combination of CRP-based task grouping, modality-specific LoRA, and intra-modality EWC is intuitive and effective. Experimental results on 16 tasks across multiple imaging modalities are strong, with clear gains in both Dice and forgetting, and the ablations help support the role of each component.

### Cons
While the proposed semantic-modality-aware continual learning framework is interesting, the experimental comparison could be further strengthened by including stronger medical image segmentation baselines, such as
- MedSAM (Segment Anything in Medical Images, Nature Communications, 2024),

to better contextualize the gains over more competitive foundation-style segmentation models. In addition, the discussion of related work on continual medical segmentation remains somewhat limited; comparing with or at least positioning against recent methods such as:

- FR²Seg (Continual Segmentation Across Multiple Sites via Fourier Adaptation AAAI 2025)

- Low-Rank Mixture-of-Experts for Continual Medical Image Segmentation (MICCAI 2024)

would make the empirical and methodological contribution clearer. More broadly, the paper could also benefit from discussing recent biomedical imaging segmentation and generation advances tailored to 2D/3D settings, such as
- HFF-Net (Rethinking Brain Tumor Segmentation from the Frequency Domain Perspective, IEEE TMI 2025), and
- TRACE (Temporally Reliable Anatomically-Conditioned 3D CT Generation with Enhanced Efficiency, MICCAI 2025),

which may provide useful context on structure-aware modeling in medical imaging.

---

> ### Author Rebuttal · Authors · 2026-03-27
>
> We sincerely thank the reviewer for these insightful suggestions and constructive feedback.
>
> **Related Work Discussion**
>
> **MedSAM** uses spatial prompts (points/bounding boxes) rather than text prompts. Our CRP framework relies on clinical text prompts to discover semantic modalities, so the two are architecturally different. Additionally, MedSAM focuses on general segmentation capability, while we address catastrophic forgetting in continual learning. Our CRP framework could potentially be extended to MedSAM by discovering modalities from spatial prompt patterns. We will explore this direction in future work.
>
> **Low-Rank MoE** assigns each task its own independent LoRA expert. For task $T$, the forward pass is $h = W_0 x + \sum_{t=1}^{T-1} B_t A_t x + B_T A_T x$, where previous experts are frozen and only the new expert $B_T A_T$ is trained. While this avoids forgetting, it has two critical limitations: (1) parameters grow linearly with task count, and (2) related tasks cannot share knowledge. For example, our benchmark contains 5 polyp segmentation datasets (Kvasir, ClinicDB, ETIS, CVC-300, ColonDB). LR-MoE creates 5 separate experts despite their semantic similarity. In contrast, our CRP automatically discovers that these tasks belong to the same modality and assigns them a single shared expert, enabling positive knowledge transfer while achieving 2.1$\times$ parameter reduction.
>
> | Method | #Experts/Modalities | Total Trainable Params |
> |:--|:--|:--|
> | LR-MoE | 16 (one per task) | 18.4M |
> | Ours | 5 (one per modality) | 8.56M |
> ||||
>
>
> **FR$^2$Seg** addresses domain shift when the same segmentation task is performed at different hospitals. It requires storing 6 complete backbone models for its adaptive transferability weighting mechanism, resulting in significant storage overhead. Our method handles a different setting—learning different segmentation tasks sequentially—with only 1 backbone + 5 lightweight LoRA modules, and is completely replay-free.
>
> | Method | Stored Models |
> |:--|:--|
> | FR$^2$Seg | 6 × backbone |
> | LR-MoE | 1 backbone + 16 LoRA |
> | Ours | 1 backbone + 5 LoRA |
> |||
>
> **HFF-Net** and **TRACE** represent important recent advances in frequency-domain segmentation and 3D CT generation respectively. We thank the reviewer for pointing out these works, which inspire potential extensions of our framework to 3D volumetric segmentation. We will discuss these promising directions in the revised paper.
>
>
> We will add detailed discussions of all these methods in the revised related work.
>
> **Key Question: Text-Guided Proxies vs. True Visual Modalities**
>
> Our discovered groups should be understood as *text-guided proxies* for task affinity, and this is by design, not a limitation.
>
> We conducted systematic experiments comparing text-based vs. visual-based clustering (please kindly see our response to Reviewer zZh1 Q1: Clustering Robustness & Visual Verification). Text embeddings provide much stronger discriminative signal: the intra-/inter-group similarity gap is $0.50+$ for text, but only $0.22$ for visual.
>
> The critical case is cardiac vs. breast ultrasound: visually almost indistinguishable (similarity $0.782$), but textually clearly separable (similarity $0.591$). Clinical text prompts naturally encode modality and anatomical information as explicit semantic content. Visual features must infer such distinctions implicitly from pixels, which is a fundamentally harder task.
>
> In short, text-guided grouping does not recover "true visual modalities" in a physical sense, but it provides a more effective structure for parameter sharing in continual learning, which is what matters for our task.

---

> > ### Author Rebuttal · Reviewer_KSx4 · 2026-04-02
> >
> > The authors’ rebuttal addressed my main concerns effectively, which slightly increased my confidence in the paper and led me to raise my score.

---

> > > ### Author Response · Authors · 2026-04-02
> > >
> > > Thank you very much for reading our rebuttal and for updating your assessment. We are glad that our clarifications helped address your concerns. We will incorporate your valuable suggestions on expanding the related work discussion in the revised version.

---

### Official Review · Reviewer_NqrU · 2026-03-11

**Soundness:** 3
**Presentation:** 2
**Significance:** 3
**Originality:** 2
**Overall Recommendation:** 4
**Confidence:** 2

**Summary:**

The paper proposes MedCRP-CL, a continual learning framework for medical image segmentation that dynamically infers task groupings from clinical text prompts without requiring predefined labels or expert counts. The method trains semantic modality-specific LoRA adapters attached to a frozen vision-language backbone and applies intra-modality EWC to prevent catastrophic forgetting among tasks assigned to the same cluster. The framework is compared against several continual learning baselines.

**Compliance With Llm Reviewing Policy:**

Affirmed.

**Final Justification:**

Since the author provided a more detailed response, I am re-evaluating the score to 4.

**Key Questions For Authors:**

- The modality discovery mechanism relies entirely on CLIP text encoder embeddings, without incorporating image features. Is there evidence that text-based modality assignments are optimal for parameter sharing? Can the authors provide a comparison between the CRP-discovered text-based groupings and groupings discovered from image features in terms of their downstream continual learning performance?

- The adaptive similarity distributions require a minimum of one intra-modality and one inter-modality observation before the Gaussian model is used. How many tasks are processed before the Gaussian estimates are sufficiently reliable? Do errors in early assignments propagate through centroid updates to affect downstream modality coherence?

- How would MedCRP-CL handle a scenario where a sequence of tasks has identical text prompts but severe visual acquisition domain shifts?

- Would splitting the Chest X-ray tasks into two semantic sub-modalities reduce the forgetting heterogeneity?

**Limitations:**

yes

**Strengths And Weaknesses:**

Strengths

- The premise of using clinical text prompts to guide task clustering is elegant.

- The model achieves a high level of parameter efficiency. The method utilizes fewer parameters than the MoE-Adapters baseline while maintaining competitive or superior average performance.

Weaknesses

- The paper does not report how many tasks are required before the Gaussian estimates become reliable, nor whether the early assignments are more error-prone. Given that early errors may propagate, it has caused me concern.

- The paper reports robustness for $\alpha >= 1$ but notes that $\alpha < 1$ causes modality collapse. But the paper does not report what modality structure is discovered for different $\alpha$ within the robust range . This is important because if the discovered modality count depends on $\alpha$, the comparison in Table 6 may partially reflect the $\alpha$ hyperparameter rather than the CRP's discovery.

- Zhang et al. (2022a) and Zhang et al. (2022b) appear to be the same paper.

---

> ### Author Rebuttal · Authors · 2026-03-27
>
> We sincerely thank the reviewer for the thoughtful comments. Below we address each concern.
>
> **Q1: Text vs Visual Grouping**
>
> Please kindly see our response to Reviewer zZh1 (Q1: Clustering Robustness & Visual Verification) for detailed analysis.
>
> **Q2: Gaussian Estimate Reliability & Error Propagation**
>
> **Early-stage safeguards.** Our framework explicitly handles unreliable early estimates through a two-stage mechanism:
> 1. *Before Gaussian activation*: We use a parameter-free logit-based likelihood, so early decisions rely solely on similarity magnitude.
> 2. *After Gaussian activation* (≥2 tasks per cluster): We begin recording intra-modality similarity statistics. The Gaussian model typically activates after 2–3 tasks and stabilizes as more observations accumulate.
>
> **Error propagation is self-correcting.** The CRP prior grows with cluster size, so larger clusters attract subsequent tasks and naturally dilute early mistakes. One misassigned task shifts the centroid by at most $\frac{1}{n}$, and this fraction shrinks as more tasks join.
>
> **Empirical validation.** Across 4 different task orderings, we obtain identical final structure ($K=5$), confirming that early variations do not affect downstream modality coherence.
>
> **Q3: Identical Prompts + Domain Shift**
>
> When tasks share identical prompts but differ in acquisition domain (e.g., chest X-ray from different hospitals), CRP assigns them to the same semantic modality. This is intended—they are the same clinical task and should share representations. Domain variation within a modality is handled by intra-modality EWC, which is exactly what it was designed for.
>
> **Q4: Chest X-ray Sub-modality**
>
> This is an interesting suggestion. However, we believe our framework can address this through $\alpha$ tuning rather than manual splitting. The CRP concentration parameter $\alpha$ directly controls clustering granularity—lower $\alpha$ encourages joining existing clusters, while higher $\alpha$ promotes finer-grained discovery. If meaningful sub-structure exists within Chest X-ray, decreasing $\alpha$ would allow CRP to automatically discover such divisions.
>
> **W: $\alpha$ Sensitivity**
>
> Please kindly see our response to Reviewer zZh1 (Q1: Clustering Robustness & Visual Verification) for detailed analysis. We report discovered $K$ for $\alpha \in \{1, 2, 5, 10\}$ in our sensitivity analysis.
>
> **W: Reference Typo**
>
> Thank you very much for catching this error! We will fix it in the revised paper.

---

> > ### Author Rebuttal · Reviewer_NqrU · 2026-04-03
> >
> > 1. The specific task threshold for Gaussian estimation to be "sufficiently reliable" remains unclear, and there is no response regarding whether early assignments are more prone to error.
> > 2. The response does not explicitly state whether the performance improvement in Table 6 comes from the method itself or from the α hyperparameter, nor does it provide details on the specific differences in modal structures under different α values.
> > 3. There is no experimental evidence to support the claim that α-modulation can automatically detect chest radiograph modalities and reduce forgetting heterogeneity; it remains only at the theoretical level.
> >
> > Since my question was deliberately avoided, I change the score to 3.
> >
> > If the concerns above can be fully addressed, I will raise my score to 4.

---

> > > ### Author Response · Authors · 2026-04-05
> > >
> > > We apologize for the insufficient detail previously.
> > >
> > > **Q1 Gaussian Estimate Reliability & Error Propagation**
> > >
> > > **Convergence under diverse initializations.** All four orderings start with different tasks (cardiac, breast, mixed, chest X-ray) yet converge to identical K=5 with identical membership. If early errors propagated, different initializations would lead to divergent structures.
> > >
> > > **Per-task assignment trace during cold-start:**
> > >
> > > | Order | Task 1 | Task 2 | Task 3 | Task 4 |
> > > |---|---|---|---|---|
> > > | grouped | kvasir NEW 0.474 | clinicdb JOIN(M1) 0.991 | etis JOIN(M1) 0.980 | cvc300 JOIN(M1) 0.992 |
> > > | interleaved | kvasir NEW 0.474 | isic NEW 0.548 | chex\_op. NEW 0.488 | busi\_ben. NEW 0.568 |
> > > | mixed | kvasir NEW 0.305 | camus NEW 0.488 | busi\_ben. NEW 0.568 | clinicdb JOIN(M1) 0.991 |
> > > | reversed | busi\_ben. JOIN(M0) 0.915 | chex\_sup. NEW 0.493 | chex\_enl. JOIN(M1) 0.790 | chex\_edema JOIN(M1) 0.870 |
> > >
> > > The cold-start logit mechanism uses 0.5 as a natural decision boundary. Same-modality pairs consistently produce sim > 0.79 while cross-modality pairs produce sim < 0.57, providing sufficient margin for reliable separation without Gaussian estimation.
> > >
> > > **Are early assignments more error-prone? No.** Gaussian first activates at task 5–9 depending on ordering, but all pre-Gaussian decisions across all 4 orderings are consistent with the final converged K=5 structure, meaning zero early assignment errors. The cold-start logit mechanism provides a 0.22+ similarity margin (same-modality > 0.79, cross-modality < 0.57), sufficient for correct separation without Gaussian estimation.
> > >
> > > **Gaussian stabilization.** The estimates become reliable within 2–3 observations after activation, with intra/inter gap remaining above 0.45 throughout:
> > >
> > > | intra n | intra\_mean | intra\_std | inter\_mean | gap |
> > > |---|---|---|---|---|
> > > | 1 | 0.980 | 0.100 (default) | 0.445 | 0.535 |
> > > | 2 | 0.925 | 0.078 | 0.448 | 0.477 |
> > > | 3 | 0.947 | 0.067 | 0.444 | 0.503 |
> > > | 8 | 0.921 | 0.064 | 0.429 | 0.492 |
> > >
> > > Final distributions are consistent across all 4 orderings (intra\_mean: 0.921–0.931, inter\_mean: 0.429–0.473), confirming that different early-stage paths do not cause divergence. Additionally, 10 text encoders spanning 4 training paradigms all discover identical K=5 with identical membership (see our response to Reviewer rzx4).
> > >
> > > **Q2 α Sensitivity**
> > >
> > > We report the discovered K and modality structure for all α values in the robust range: α=2, 5, 7, 10 all produce identical K=5 with identical structure: {cardiac}, {polyp×5}, {dermoscopy}, {chest×7}, {breast×2}.
> > > Since our framework deterministically maps each task to a LoRA adapter based on its modality assignment, identical structure implies identical training dynamics. The performance improvement in Table 6 therefore comes entirely from the method's ability to discover a stable 5-modality semantic structure (separating cardiac and breast ultrasound), not from the specific choice of α.
> > >
> > > **Q3 Chest X-ray Sub-modality via α Tuning**
> > >
> > > **α does split chest X-ray at higher values, but affects all modalities simultaneously.** Within the robust range (α > 1), CRP consistently discovers K=5, and all 7 chest X-ray tasks are always grouped together. Beyond α=10, CRP fragments all modalities toward K=16 (one adapter per task). Finer granularity does reduce forgetting, but it also eliminates positive transfer within groups. α controls global clustering granularity and cannot selectively refine one modality while keeping others intact. Any α large enough to split chest X-ray also fragments the 5 polyp datasets into separate clusters.
> > >
> > > **Manual splitting does reduce forgetting heterogeneity.** We conducted experiments with two clinically motivated grouping criteria, each splitting the 7 chest X-ray tasks into two sub-modalities (K=6 total):
> > >
> > > Split A (anatomical region): pulmonary (Opacity, Atelectasis, Edema, Effusion) vs. mediastinal/device (Cardiomegaly, Enl. Cardio., Support Dev.).
> > >
> > > Split B (cardiac relevance): cardiac-related (Cardiomegaly, Enl. Cardio., Edema) vs. non-cardiac (Opacity, Atelectasis, Effusion, Support Dev.).
> > >
> > > | Setting | Chest Fgt std | Chest Dice | Overall Dice |
> > > |---|---|---|---|
> > > | K=5 (CRP) | 0.041 | 0.650 | 0.733 |
> > > | K=6 Split A | 0.021 | 0.683 | 0.768 |
> > > | K=6 Split B | 0.020 | 0.671 | 0.758 |
> > >
> > > Both splits substantially reduce forgetting heterogeneity (std from 0.041 to ~0.021). The consistent improvement across two different clinical criteria suggests that the benefit comes from increased parameter isolation, not from any specific grouping choice.
> > >
> > > **Why we do not adopt K=6.** Both splits require manual clinical knowledge, which is inconsistent with our framework's core design of automatic modality discovery. The 7 chest X-ray tasks share text embedding similarity >0.83, leaving no separable sub-cluster boundary for text-based CRP. Automatic discovery of such sub-structure would require hierarchical CRP or visual/acquisition metadata as secondary clustering signals.

---

### Official Review · Reviewer_rzx4 · 2026-03-12

**Soundness:** 2
**Presentation:** 3
**Significance:** 3
**Originality:** 4
**Overall Recommendation:** 4
**Confidence:** 2

**Summary:**

This paper proposes MedCRP-CL, a continual learning framework for medical image segmentation that addresses the challenge of sequentially learning from heterogeneous imaging tasks while mitigating catastrophic forgetting. The method performs online task structure discovery using a Chinese Restaurant Process (CRP) to cluster tasks into semantic modalities based on clinical text prompts. The model builds on a frozen CLIPSeg backbone with modality-specific LoRA adapters, and applies intra-modality Elastic Weight Consolidation (EWC) to preserve knowledge within each cluster. Experiments across 16 segmentation tasks spanning four imaging modalities demonstrate strong performance, achieving 73.3% average Dice, outperforming prior methods while maintaining low forgetting and improved parameter efficiency.

**Compliance With Llm Reviewing Policy:**

Affirmed.

**Final Justification:**

The authors has addressed all my questions, I decide to raise the score.

**Key Questions For Authors:**

1. How robust is the proposed clustering mechanism to noisy, ambiguous, or inconsistent clinical prompts?
2. As new semantic modalities are discovered, additional LoRA adapters are allocated. How does the method scale in long continual learning scenarios with many tasks, and is there a mechanism to control adapter growth?
3. Several datasets used in the experiments contain very few training samples, raises question about the validity of the improved performance.

**Limitations:**

yes

**Strengths And Weaknesses:**

The paper presents a well-motivated framework for continual medical image segmentation and supports its claims with comprehensive experiments across 16 tasks spanning multiple imaging modalities. The methodology is coherent, combining CRP-based task clustering with LoRA adapters and EWC regularization to balance parameter sharing and isolation. The authors provide theoretical justification for the clustering mechanism and include thorough ablation studies to validate the contributions of key components. The paper is generally well organized, with clear motivation and helpful figures illustrating the architecture and training pipeline. The work is also practically relevant: the replay-free design aligns with clinical privacy constraints, and the method achieves strong performance while maintaining high parameter efficiency. The idea of clustering tasks based on clinical text prompts rather than image features is an interesting and intuitive perspective for task structure discovery.

However, some datasets used in the evaluation are extremely small, which may introduce high variance and limit the reliability of the reported improvements. The clustering mechanism relies heavily on embeddings from a frozen CLIP text encoder, but the robustness of this approach to noisy or ambiguous prompts is not analyzed. Finally, the broader applicability of the approach is somewhat unclear, as the framework is designed for vision-language segmentation models and may not directly extend to unimodal or 3D medical imaging settings.

---

> ### Author Rebuttal · Authors · 2026-03-27
>
> We sincerely thank the reviewer for the thoughtful questions. Below we address each concern.
>
> **Q1: Clustering Robustness**
>
> Please kindly see our response to Reviewer zZh1 (Q1: Clustering Robustness & Visual Verification) for detailed analysis.
>
> **Q2: Adapter Scalability / Growth Control**
>
> While CRP currently lacks an explicit merging mechanism, we note that modality growth is naturally bounded in practice:
> - **Modality growth is sublinear.** With 16 tasks spanning 4 physical imaging modalities, CRP discovers only $K=5$ semantic modalities. Most new tasks join existing clusters rather than creating new ones.
> - **Each LoRA adapter adds approximately 0.2M parameters.** Even with $K=10$ modalities, the total overhead remains under 2M parameters, less than 0.5% of the backbone.
> - **Medical imaging modalities are inherently finite**, including X-ray, CT, MRI, ultrasound, endoscopy, dermoscopy, and others. We expect $K$ to saturate at a manageable level rather than grow indefinitely.
>
> We plan to explore adapter merging when two modalities' cluster centroids become sufficiently similar (e.g., cosine similarity $> 0.85$), using Fisher-weighted averaging to consolidate LoRA parameters while preserving learned knowledge.
>
> **Q3: Small Dataset Validity**
>
> We thank the reviewer for this important concern. We address the validity of improvements on small datasets from multiple perspectives:
>
> In real-world clinical deployment, specialized imaging and rare pathologies often result in extremely limited annotated data. For instance, findings such as Edema ($n=25$) and Cardiomegaly ($n=35$) represent the genuine scarcity of expert-labeled samples in chest X-ray analysis. Our method is specifically designed for these small-sample challenges within a continual learning framework.
>
> We report standard deviations across 4 task orderings (Table 2: $73.3\% \pm 1.0\%$ Dice, $4.1\% \pm 0.8\%$ forgetting). All baselines are evaluated on identical datasets. All methods face the same small-sample challenges. This lower variance indicates robust performance, regardless of where these small-sample tasks appear in the learning sequence.
>
> **Limitation: Applicability to Unimodal / 3D Settings**
>
> Our current work focuses on 2D vision-language segmentation. As clinical text prompts provide rich semantic signals that enable effective task clustering, which is the core contribution we aim to validate.
>
> Extension to 3D volumetric imaging is an interesting direction. Alternative task descriptors (e.g., DICOM metadata, learned visual prototypes) could potentially serve this role. We will clarify the current scope in the revised paper and discuss these extensions as future directions.

---

> > ### Author Rebuttal · Reviewer_rzx4 · 2026-04-02
> >
> > Thank you for the detailed response. I have one follow-up question about the method's sensitivity to encoder choice. How much does the performance depend on using CLIP for text and CLIPSeg for vision? Have you evaluated alternative encoders, or do you expect the method to generalize well across different encoder choices?

---

> > > ### Author Response · Authors · 2026-04-03
> > >
> > > Thank you for the excellent follow-up question. We tested 10 text encoders across 4 architectural categories to evaluate encoder sensitivity.
> > >
> > > **Setup:** We fix the 16 tasks and their ordering, replace only the text encoder used for CRP clustering, and sweep $\alpha \in \{2, 5, 7, 10\}$.
> > >
> > > | Category | Encoder | Training Paradigm | $\alpha$=2 | $\alpha$=5 | $\alpha$=7 | $\alpha$=10 |
> > > |---|---|---|---|---|---|---|
> > > | Baseline | CLIP (CLIPSeg built-in) | Contrastive | 5 | 5 | 5 | 5 |
> > > | Medical VLP | PubMedCLIP | Contrastive (medical) | 5 | 5 | 5 | 5 |
> > > | Medical VLP | BiomedCLIP | Contrastive (medical) | 1 | 2 | 5 | 5 |
> > > | Medical VLP | MedICaT-ROCO | Contrastive (medical) | 5 | 5 | 5 | 10 |
> > > | General VLP | OpenCLIP-B/32 | Contrastive (LAION) | 2 | 5 | 5 | 5 |
> > > | General VLP | OpenCLIP-B/16 | Contrastive (LAION) | 2 | 5 | 5 | 5 |
> > > | General VLP | OpenCLIP-L/14 | Contrastive (LAION) | 4 | 5 | 5 | 5 |
> > > | Non-CLIP | FLAVA | Contrastive+MLM+MIM+ITM | 4 | 5 | 5 | 5 |
> > > | Non-CLIP | BLIP | ITC+ITM+LM | 3 | 5 | 5 | 5 |
> > > | Non-CLIP | CoCa | Contrastive+Generative | 2 | 5 | 5 | 5 |
> > >
> > > **(1) All 10 encoders discover the identical $K$=5 grouping, with membership exactly matching that reported in our Q1 response to Reviewer zZh1.** These 10 encoders span fundamentally different training paradigms: contrastive-only (CLIP family), contrastive+generative (CoCa), contrastive+masked modeling (FLAVA), and contrastive+matching+language modeling (BLIP). The semantic modality structure is therefore an intrinsic property of the data, not an artifact of any specific encoder.
> > >
> > > **(2) Encoders differ only in embedding space compactness, which shifts the optimal $\alpha$ range.** Compact spaces under-segment at low $\alpha$ (e.g., BiomedCLIP requires $\alpha \geq 7$ to reach $K$=5), while fine-grained encoders over-segment at high $\alpha$ (e.g., MedICaT-ROCO yields $K$=10 at $\alpha$=10). These two medical VLP models exhibit narrower stable $\alpha$ ranges than general-purpose encoders, likely due to their more domain-specific and smaller-scale training data. This also justifies our default choice of the CLIP encoder, which maintains $K$=5 across the entire $\alpha$ range tested.
> > >
> > > **(3) The CRP clustering module is decoupled from the segmentation backbone.** It requires only text embeddings as input. Our results show it generalizes across training paradigm, model scale, and training domain.
> > >
> > > **(4) Contrast with visual-only clustering.** As shown in our Q1 response, visual-only clustering produces unstable $K$ values that shift with $\alpha$ ($K$=1 at $\alpha \leq 2$, $K$=4 at $\alpha$=10) and cannot separate cardiac from breast ultrasound (visual similarity 0.95+). In contrast, text-based CRP yields stable $K$=5 across all 10 encoders, further confirming the superiority of text prompts as the clustering signal.
> > >
> > > **Limitation on encoder requirements.** We also tested SigLIP (sigmoid-loss VLP) and S-PubMedBERT (sentence-level language model, no vision pre-training). Both produced $K$=1 across all $\alpha$ values. SigLIP optimizes absolute matching scores rather than relative ranking, resulting in an embedding space where inter-task distances are insufficiently separated for CRP clustering. S-PubMedBERT is trained for sentence-level retrieval and does not produce sufficiently fine-grained distinctions for short medical prompts. Our method requires a text encoder with adequate semantic discriminability. All 10 VLP models with InfoNCE/softmax contrastive objectives in our evaluation satisfy this requirement and recovered the identical grouping structure.
> > >
> > > We will include this encoder sensitivity analysis in the revised paper.

---

### Official Review · Reviewer_zZh1 · 2026-03-12

**Soundness:** 3
**Presentation:** 3
**Significance:** 3
**Originality:** 2
**Overall Recommendation:** 3
**Confidence:** 4

**Summary:**

To address the fundamental tension between parameter sharing and isolation in continual medical image segmentation, this paper introduces MedCRP-CL, a novel framework for online task structure discovery. The core innovation lies in leveraging the Chinese Restaurant Process (CRP) to dynamically group sequential tasks into "semantic modalities" based on clinical text prompts, rather than relying on coarse physical imaging labels. By maintaining modality-specific LoRA adapters regularized by intra-modality EWC, the framework achieves effective knowledge transfer among similar tasks while preventing catastrophic forgetting across dissimilar ones.

**Compliance With Llm Reviewing Policy:**

Affirmed.

**Key Questions For Authors:**

1.The framework’s task discovery heavily relies on the quality and specificity of clinical text prompts. In real-world scenarios, clinical metadata is often noisy, abbreviated, or even missing. How does the model handle cases where two tasks have similar text descriptions but fundamentally different visual distributions (or vice versa)? Have the authors considered incorporating visual feature consistency into the CRP likelihood term to provide a "dual-track" verification, ensuring that task grouping is robust to linguistic ambiguities?
2.While the CRP mechanism excels at discovering new modalities, it lacks a mechanism to merge existing ones. As the task sequence grows indefinitely, the number of modality-specific LoRA adapters will increase linearly, leading to substantial storage overhead and fragmented knowledge representation. Do the authors plan to introduce a merging or pruning strategy？

**Limitations:**

1.The framework emphasizes parameter isolation across different semantic modalities to prevent forgetting. However, this rigid separation may preclude beneficial knowledge transfer between distantly related modalities. 2.Although LoRA adapters are parameter-efficient individually, the total parameter count grows linearly with the number of discovered semantic modalities. In an indefinite continual learning scenario, the accumulation of numerous adapters could lead to substantial storage overhead and increased model management complexity.

**Strengths And Weaknesses:**

Soundness:Using a Bayesian Nonparametric approach (CRP) to model task relationships is theoretically sound, as it avoids the heuristic of pre-defining cluster counts. The integration of LoRA for parameter efficiency and EWC for intra-modality stability provides a multi-layered defense against catastrophic forgetting. The empirical results are extensive and include Order Sensitivity Analysis that confirms the framework's robustness.
Presentation:The submission is well-structured and follows a logical progression from clinical motivation to mathematical formulation. The distinction between "Physical Modality" and "Semantic Modality" is clearly articulated, which is vital for understanding the paper's core contribution. However, the explanation of the likelihood term that modulates the CRP prior could be more detailed to ensure full reproducibility.
Significance:This work allows a model to be deployed in a hospital and autonomously figure out how to categorize incoming data streams without constant manual re-labeling or architectural adjustments.
Originality:While MedCRP-CL demonstrates impressive empirical results on a wide range of tasks, the conceptual novelty is somewhat limited. The framework primarily re-packages existing methodologies—specifically CRP for task discovery and LoRA/EWC for adaptation—into a medical segmentation pipeline. The core insight—grouping tasks by prompt similarity—is an incremental evolution of prompt-based learning rather than a transformative new perspective.

---

> ### Author Rebuttal · Authors · 2026-03-27
>
> We sincerely thank the reviewer for the constructive feedback. Below we address each concern.
>
> **Q1: Clustering Robustness & Visual Verification**
>
> We conducted experiments comparing text-only, visual-only, and dual-track (50% text + 50% visual) clustering. Both embeddings are extracted from the same CLIP backbone within CLIPSeg.
>
> **Pairwise similarity analysis across 16 tasks:**
>
> | Embedding | Inter-group mean | Intra-group mean | Min |
> |-----------|------------------|------------------|-------|
> | Visual | 0.727 | 0.95+ | 0.522 |
> | Text | 0.45 | 0.95+ | 0.336 |
>
> Text embeddings provide stronger discriminative signal—the intra-/inter-group similarity gap is $0.50+$, compared to just $0.22$ for visual.
>
> **Discovered $K$ across $\alpha$ and strategies:**
>
> | $\alpha$ | Text-only | Visual-only | Dual-track (50/50) |
> |----------|-----------|-------------|---------------------|
> | 1 | 5 | 1 | 1 |
> | 2 | 5 | 1 | 1 |
> | 5 | 5 | 1 | 4 |
> | 10 | 5 | 4 | 5 |
>
> Under CRP, $\mathbb{E}[K] = \sum_{i=1}^{N} \frac{\alpha}{\alpha+i-1}$. For $N=16$: $\mathbb{E}[K] \approx 1.9$ ($\alpha=0.5$), $3.4$ ($\alpha=1$), $5.8$ ($\alpha=2$), $9.2$ ($\alpha=5$), $12.1$ ($\alpha=10$). The range $\alpha \in [1,10]$ avoids both extremes—$\alpha < 1$ forces under-clustering ($\mathbb{E}[K] < 2$), while $\alpha > 10$ approaches one-per-task partitions ($\mathbb{E}[K] \to N$).
>
> Text-only discovers stable $K=5$ across all $\alpha$; visual-only and dual-track produce inconsistent $K$ that shifts with $\alpha$. At $\alpha=5$, dual-track yields $K=4$ (merging cardiac and breast ultrasound), identical to physical modality grouping (Table 6: Dice=65.75%, Fgt=9.23%). Adding visual features degrades the correct $K=5$ structure that text-only achieves.
>
> The critical case: cardiac vs. breast ultrasound has visual similarity $0.782$ (indistinguishable) but text similarity $0.591$ (clearly separable). Clinical prompts explicitly name the modality and anatomy, while visual features must infer the same from pixels alone.
>
> **Robustness to noisy prompts:**
>
> *Original:* "Left ventricular cavity of rectangle shape in two-chamber view of the heart at end of the diastole cycle of a 56-year-old f with good image quality."
>
> | Perturbation | Level | $K$ | Example |
> |--------------|-------|-----|---------|
> | **Clinically Realistic** | | | |
> | Abbreviation | - | 5 | ...in 2CV...at ED...with good IQ. |
> | Realistic typo | 10% | 5 | ...two-chamber veiw...image qualty. |
> | Realistic typo | 20% | 5 | ...venricular cavit...sshape...two-chambr vie... |
> | Keyword drop | 20% | 5 | ...two-chamber view the heart end of the cycle... |
> | Keyword drop | 30% | 5 | ...rectangle two-chamber view the heart end of the cycle... |
> | Shuffle | - | 5 | diastole at of quality. a in shape the the... |
> | **Extreme Perturbations** | | | |
> | Realistic typo | 30% | 3 | ...tthe heart at ennd of tge diastle... |
> | Keyword drop | 50% | 1 | Left cavity of rectangle two-chamber view... |
> | Generic | - | 1 | segment the region |
>
> Under clinically realistic perturbations (abbreviations, ≤20% typos, ≤30% keyword drop, word reordering), CRP discovers the same $K=5$ with 100% consistency. Degradation under extreme perturbations is principled—when prompts lose discriminability, CRP defaults to increased parameter sharing.
>
> **Q2 & Limitation 2: Modality Merging and Storage Overhead**
>
> Please kindly see our response to Reviewer zzx4 (Q2: Adapter Scalability / Growth Control).
>
> **Limitation 1: Cross-Modality Knowledge Transfer**
>
> We want to clarify that knowledge transfer does happen at two levels in our framework:
>
> - **Backbone sharing.** The frozen CLIPSeg backbone (151M parameters) is shared across all tasks. We isolate only the LoRA adapters (0.2M each)—a small fraction of total parameters.
> - **Intra-modality transfer.** Within each semantic modality, tasks share the same LoRA adapter. This is where intra-modality transfer happens. For example, when multiple chest X-ray tasks are assigned to the same modality, later tasks benefit from earlier learning. Intra-modality EWC ensures this transfer happens without forgetting.
>
> **Regarding Originality:**
>
> We appreciate the opportunity to clarify. Our contribution is a new problem formulation, not the individual modules. The key question we address: at what granularity should tasks be grouped for parameter sharing in medical continual learning? Prior work groups tasks by physical imaging modality, but this is inadequate—cardiac and breast ultrasound are visually similar yet clinically distinct.
>
> We term the correct granularity semantic modality and show that clinical text prompts naturally encode this structure without additional annotation. CRP enables online discovery without predefining the number of groups. The framework is also replay-free, aligning with clinical privacy requirements.
>
> This finer-grained structure enables 7.6% higher Dice and 5% lower forgetting versus physical modality grouping (Table 6). We will revise to clarify these contributions.

---

> > ### Author Rebuttal · Reviewer_zZh1 · 2026-04-05
> >
> > Thanks for the additional experiments and detailed response.  While the use of individual LoRA adapters ($0.2$M) ensures parameter efficiency, how does the framework prevent 'knowledge fragmentation' in a truly indefinite deployment scenario? Specifically, without a principled merging or pruning mechanism, the model appears to only accumulate disjoint modalities. How can the system consolidate overlapping clinical insights that may exist across similar but separately discovered semantic clusters as the task sequence grows significantly?

---

> > > ### Author Response · Authors · 2026-04-06
> > >
> > > Thank you for the follow-up question. We address the knowledge fragmentation concern with comprehensive experimental validation.
> > >
> > > **Experimental setup.** We use Fisher-weighted averaging to merge two modalities' LoRA adapters, where the Fisher information matrices serve as weights and are already computed during EWC. After merging, we re-adapt all affected tasks for 5 epochs by tuning the task adapters, seg heads, and enhancers while freezing the merged LoRA. Training loss plateaus by epoch 3–4 in all cases. We exhaustively test all 10 cross-modality merge pairs.
> > >
> > > **Results:**
> > >
> > > | Merge Pair | Modalities | Before | After | $\Delta$ Affected |
> > > |---|---|---|---|---|
> > > | M0+M1 | Cardiac+Polyp | 0.793 | 0.398 | $-$0.395 |
> > > | M0+M2 | Cardiac+Dermoscopy | 0.871 | 0.643 | $-$0.228 |
> > > | M0+M3 | Cardiac+ChestXR | 0.686 | 0.538 | $-$0.148 |
> > > | M0+M4 | Cardiac+BreastUS | 0.772 | 0.457 | $-$0.315 |
> > > | M1+M2 | Polyp+Dermoscopy | 0.812 | 0.654 | $-$0.158 |
> > > | M1+M3 | Polyp+ChestXR | 0.718 | 0.544 | $-$0.174 |
> > > | M1+M4 | Polyp+BreastUS | 0.778 | 0.546 | $-$0.232 |
> > > | M2+M3 | Dermoscopy+ChestXR | 0.701 | 0.685 | $-$0.016 |
> > > | M2+M4 | Dermoscopy+BreastUS | 0.812 | 0.754 | $-$0.058 |
> > > | M3+M4 | ChestXR+BreastUS | 0.687 | 0.647 | $-$0.039 |
> > >
> > > All 10 merge pairs degrade performance ($\Delta$ from $-$0.016 to $-$0.395). This holds despite using Fisher-weighted merging, a principled parameter consolidation method widely adopted in continual learning, and despite allowing re-adaptation to convergence. We also verified that the least harmful merge (M2+M3, $\Delta$=$-$0.016) is consistently negative across all task orderings, confirming that this small drop is not within noise.
> > >
> > > **Analysis.** This exhaustive experiment demonstrates that each of the 5 discovered semantic modalities captures genuinely distinct feature representations rather than redundant fragments of shared knowledge. When we merge any two modalities' adapters, the resulting mutual interference cannot be recovered through re-adaptation. The degradation is also broadly correlated with modality dissimilarity: similar pairs such as M2+M3 and M3+M4 suffer least, while dissimilar pairs such as M0+M1 and M0+M4 suffer most. This pattern confirms that CRP has correctly identified the task structure, and that modalities CRP separates are better kept separate.
> > >
> > > Regarding the reviewer's concern about indefinite deployment: in our current 16-task setting across 4 physical imaging modalities, CRP discovers 5 semantic modalities with no overlapping clusters. If future tasks were to create a partially overlapping cluster, the merge degradation metric we demonstrate here could serve as a principled criterion. Specifically, adapter consolidation should only proceed when the merge cost ($\Delta$) is negligible.
> > >
> > > We also emphasize that cross-modality knowledge sharing already occurs through the frozen backbone. The CLIPSeg backbone contains 151M parameters (99.87% of total) and is fully shared across all tasks, encoding generalizable visual-language features. Each LoRA adapter adds only 0.2M parameters (0.13%) and captures modality-specific residual adjustments. The "fragmentation" thus affects only this minimal fraction of parameters, and these parameters are inherently modality-specific by design.
> > >
> > > We will include this merge analysis in the revised paper.

---

### Decision · Program_Chairs · 2026-04-30

**Decision:**

Accept (regular)

**Comment:**

This paper initially received mixed reviews. The rebuttal addressed most of concerns raised by the reviewers. Except one reviewer (zZh1) lands on the negative side, the other three reviewers all recommended acceptance of the paper. The remaining concern raised by Reviewer zZh1, i.e., knowledge fragmentation, is mostly caused by a truly indefinite deployment scenario, which is somewhat extreme. Thus, the AC lean to accept the paper.